# Small Non-Coding-RNA in Gynecological Malignancies

**DOI:** 10.3390/cancers13051085

**Published:** 2021-03-03

**Authors:** Shailendra Kumar Dhar Dwivedi, Geeta Rao, Anindya Dey, Priyabrata Mukherjee, Jonathan D. Wren, Resham Bhattacharya

**Affiliations:** 1Department of Obstetrics and Gynecology, University of Oklahoma Health Sciences Center, Oklahoma City, OK 73104, USA; shailendra-dwivedi@ouhsc.edu (S.K.D.D.); Anindya-Dey@ouhsc.edu (A.D.); 2Department of Pathology, University of Oklahoma Health Sciences Center, Oklahoma City, OK 73104, USA; Geeta-Rao@ouhsc.edu (G.R.); Priyabrata-Mukherjee@ouhsc.edu (P.M.); 3Peggy and Charles Stephenson Cancer Center, University of Oklahoma Health Sciences Center, Oklahoma City, OK 73104, USA; 4Biochemistry and Molecular Biology Department, University of Oklahoma Health Sciences Center, Oklahoma City, OK 73104, USA; Jonathan-Wren@omrf.org; 5Oklahoma Center for Neuroscience, University of Oklahoma Health Sciences Center, Oklahoma City, OK 73104, USA; 6Genes & Human Disease Research Program, Oklahoma Medical Research Foundation, Oklahoma City, OK 73104, USA; 7Department of Cell Biology, University of Oklahoma Health Science Center, Oklahoma City, OK 73104, USA

**Keywords:** small non-coding-RNA, gynecological malignancies, microRNAs (miRs), P-Element induced wimpy testis interacting (PIWI) RNAs (piRNAs), tRNA-derived small RNAs, ovarian cancer, endometrial cancer, cervical cancer

## Abstract

**Simple Summary:**

Gynecologic malignancies are among the leading cause of female mortality worldwide, and their management is complicated by late diagnosis and acquired therapy resistance. Although altered DNA code, leading to aberrant protein expression, is indispensable for cancer initiation and progression, from the current literature it is clear that, not only proteins, but also noncoding RNA, which does not translate into proteins, can also be instrumental. Based on their size, noncoding RNA, are further classified into long and small noncoding RNA. Here, we have comprehensively reviewed the literature about the role of small noncoding RNAs in gynecological malignancies, and discussed how these small noncoding RNA can be vital for diagnosis and therapy.

**Abstract:**

Gynecologic malignancies, which include cancers of the cervix, ovary, uterus, vulva, vagina, and fallopian tube, are among the leading causes of female mortality worldwide, with the most prevalent being endometrial, ovarian, and cervical cancer. Gynecologic malignancies are complex, heterogeneous diseases, and despite extensive research efforts, the molecular mechanisms underlying their development and pathology remain largely unclear. Currently, mechanistic and therapeutic research in cancer is largely focused on protein targets that are encoded by about 1% of the human genome. Our current understanding of 99% of the genome, which includes noncoding RNA, is limited. The discovery of tens of thousands of noncoding RNAs (ncRNAs), possessing either structural or regulatory functions, has fundamentally altered our understanding of genetics, physiology, pathophysiology, and disease treatment as they relate to gynecologic malignancies. In recent years, it has become clear that ncRNAs are relatively stable, and can serve as biomarkers for cancer diagnosis and prognosis, as well as guide therapy choices. Here we discuss the role of small non-coding RNAs, i.e., microRNAs (miRs), P-Element induced wimpy testis interacting (PIWI) RNAs (piRNAs), and tRNA-derived small RNAs in gynecological malignancies, specifically focusing on ovarian, endometrial, and cervical cancer.

## 1. Introduction

### 1.1. Gynecologic Cancers

In the United States, in 2020, gynecologic cancers constituted ~6% of all cancer patients, accounting for ~5.3% of all cancer related deaths. Typically, the primary, and most effective, treatment for these cancers is optimal surgery; however, surgery is effective only if the disease is diagnosed at an early stage. Due to recent clinical advances, including the availability of effective screening tools, both endometrial and cervical cancer are increasingly likely to be diagnosed at an early stage. Disappointingly, ovarian cancer is designated the “silent killer”, because it is often diagnosed at an advanced stage where curative treatment is difficult; treatment of ovarian cancer is often further challenged by recurrence and acquired drug resistance. These malignancies have been reviewed in detail elsewhere [1,2,3,4], and a brief summary is presented in the following sections:

#### 1.1.1. Cervical Cancer

Cervical cancer is the fourth most commonly diagnosed malignancy worldwide [5], constituting approximately 50% of all malignancies of the female reproductive system. Cervical cancer is most often diagnosed in women between the ages of 35 and 44 years, although the average age at diagnosis is 50 years. Overall, ~85% of new cases, as well as deaths, occur in developing countries. In the United States over the last twenty years the case rate has fallen from 8.9 to 6.6 per 100,000 population, mainly due to increased testing. The most common patho-histological form of cervical cancer is planocellular (squamous cell carcinoma), which represents over 90% of all cancers at this site [6]. Approximately 6% are adenocarcinomas and less than 2% are adenosquamous (mixed) carcinomas [7]; the remaining are rare carcinomas and cervical sarcomas [8]. Based on the degree of differentiation, cervical cancers are divided into, unknown (g_x_), good (g_1_), medium (g_2_), poor (g_3_), and undifferentiated (g_4_) grades. Hematogenic metastasis is relatively rare and occurs quite late, such that cervical cancer remains a pelvic disease for a prolonged period [9]; the most common sites for metastasis are the lungs and the liver. Bone metastases are rare, and when found they are usually in the vertebrae of the spinal column or, even more rarely, the long bones of the lower extremities. Most cervical cancer cases are caused by the sexually transmitted human papillomavirus (HPV), which is present in more than 90% of tumors [10]. Harald zur Hausen identified HPV in cervical cancer patients and, received the Nobel Prize in Physiology or Medicine in 2008 for his study of the etiology of cervical cancer and the role of HPV in its genesis [11]. The HPV-E6 protein complexes with cellular proteins, ubiquitin-protein ligase E3A (E6AP), and p53, facilitating p53 degradation via the ubiquitin dependent proteolytic system [12], and leading to invasive cervical cancer.

#### 1.1.2. Endometrial Cancer

More than 90% of uterine cancers occur in the endometrium, and endometrial cancer is the most commonly diagnosed gynecologic malignancy in developed countries [13]. In the USA, a projected 65,620 patients will be diagnosed with, and approximately 12,590 women will die from, endometrial cancer in 2020. The number of women diagnosed with endometrial cancer is increasing, largely due to increased obesity rates; obesity is an important risk factor for endometrial cancer. From 2007 to 2016, the number of white women diagnosed with endometrial cancer increased annually by 1%, while in the same timeframe diagnoses in black women increased by 2%. Nevertheless, 67% of women with endometrial cancer are diagnosed at an early, and more readily treatable, stage. Most endometrial cancers are sporadic; ~5% are considered hereditary and caused by mutations in the DNA mismatch repair genes [14]. From 2008 to 2017, deaths from uterine cancer increased by approximately 2%.

Traditionally, endometrial cancers are classified by histo-morphologic features, and stratified into the more common, lower risk, estrogen-driven type I cancers, and the less common, more aggressive, non-estrogen-driven type II cancers [15]. Type I endometrial cancer arises from pre-neoplastic lesion hyperplasia that has undergone unchecked estrogenic stimulation [16], and Type II carcinomas develop from atrophic endometrium and are frequently serous or clear-cell adenocarcinomas. The most comprehensive molecular study of endometrial cancer to date was provided by The Cancer Genome Atlas (TCGA) project, which included a combination of whole genome sequencing, exome sequencing, microsatellite instability (MSI), and copy number analysis [17]. This molecular information was used to classify 232 endometrioid and serous endometrial cancers into four groups, *POLE* ultra-mutated, MSI hyper mutated, copy-number (CN) low, and CN high, that correlate with progression-free survival (reviewed in detail by McAlpine [18]).

#### 1.1.3. Ovarian Cancer

Ovarian cancer remains a major cause of both morbidity and mortality in women, with little improvement in survival rates over the past four decades. Ovarian tumors are divided into three clinico-pathological subtypes with distinct histopathological features: epithelial, sex cord-stromal, and germ cell tumors. Of these, epithelial ovarian cancer (EOC) is the most common, representing 80–85% of ovarian cancers [19]. Based on histological and morphological differences, EOC is classified into five major categories: high-grade serous, low-grade serous, mucinous, endometrioid, and clear-cell carcinoma. High-grade serous ovarian cancer (HGSOC) is the most frequently seen and lethal histotype, causing nearly 75% of all EOC-related mortalities [19]. Historically, HGSOC was thought to originate from the ovarian surface epithelium; however, recent studies have indicated that the majority of advanced HGSOC may arise from the fallopian tube fimbriae [20]. Within the most common subtype, HGSOC, TP53 is mutated in over 90% of patients. Advances in next-generation sequencing have revealed that mutations within the DNA repair pathways, including BRCA1 and BRCA2, are common in about 50% of HGSOC patients; this has led to the development of a breakthrough therapy targeting poly ADP ribose polymerase (PARP) via inhibitors. These vulnerabilities of HGSOC are being explored, with recent reports indicating an interplay between *BRCA1*-mutation status and progesterone signaling. Therefore, treatment with anti-progestins could be an effective nonsurgical, prophylactic option for ovarian and breast cancer prevention in these high-risk women [21].

Current diagnostic methods for the detection and monitoring of ovarian cancer include pelvic examination, transvaginal ultrasound, and measurement of the serum biomarker carbohydrate antigen 125 (CA125) [22], human epididymis protein 4 (HE4) [23] Wnt/beta-catenin [24], and p53 [25]; these methods have certainly improved outcomes, but have limitations and lack adequate sensitivity. Ovarian cancer is asymptomatic in its early stages making early diagnosis difficult. About 75% of patients are diagnosed at stage III or IV with extensive metastasis in the peritoneal cavity, and have a 5-year survival rate of less than 40% [26]. In contrast, patients who are diagnosed at stages I or II, have 5-year survival rates of 70–90%. The high mortality rate of ovarian cancer can be partly attributed to its detection at late stages.

While current research has provided tools for improved diagnosis and management of gynecologic malignancies, the continuing socio-economic and racial disparities in gynecologic cancers highlight the need for further studies to develop sensitive diagnostics for early detection and efficacious targeted therapy with reduced toxicity. However, the potential roles of small ncRNAs in regulating the proteome via interaction with mRNA, DNA, and proteins have not been fully explored.

## 2. Small ncRNAs

The term non-coding RNA refers to a large group of endogenous RNA molecules that have no protein coding capacity. The Encyclopedia of DNA Elements (ENCODE) consortium reports that around 70% of the genome is transcribed for RNA molecules [27] that have no protein coding capacity. Rather than being mere transcriptional noise, these molecules are powerful regulators of gene expression and function as structural, catalytic, and regulatory RNAs [28,29,30,31]. Non-coding RNAs are further divided into small non-coding RNAs (sncRNAs) with sizes <200 nt (e.g., miR, piRNA, and tiRNA), and long non-coding RNAs (lncRNA) with sizes ≥200 nt (e.g., lincRNA, NAT). The defining features of sncRNAs are their short length, their association with members of the argonaute (Ago) family of proteins, and that they usually downregulate or silence target gene expression. Beyond these defining features, different sncRNA classes guide diverse and complex schemes of gene regulation [32].

The aberrant expression of sncRNAs is associated with various cellular dysfunctions and disease states. Increasing evidence suggests that multiple sncRNA groups play important roles in cancer initiation, progression, and associated pathophysiology. Clinically, sncRNA aberrations show high diagnostic and prognostic value. With improved understanding of the nature and roles of non-coding RNAs, it is believed that we can develop cancer therapeutics that act via the modulation of such RNA molecules. Advances in in-vivo nucleic acid delivery methods and in-silico approaches have prompted the development of agents that may disrupt the oncogenic functions of non-coding RNAs. In this review, we will briefly discuss the roles of piRNA, tRNA, and miR in endometrial, cervical, and ovarian cancers.

### 2.1. piRNA

P-Element induced wimpy testis interacting (PIWI) RNAs (piRNAs) are sncRNAs of approximately 24–31 nucleotides; they have a 5′-terminal uridine or tenth position adenosine bias, lack clear secondary structure motifs, and interact with PIWI, which are nuclear RNA-binding proteins. piRNAs were initially described in germline cells, but emerging evidence reveals they are expressed in a tissue-specific manner among multiple human somatic tissue types, as well as in cancer [33]. piRNAs were first identified in fly testis as a novel class of “long siRNAs” that silence Stellate, a multi-copy gene on the *Drosophila melanogaster*, X chromosome [34]. Depending on the source, piRNAs are divided into three groups: lncRNA-derived piRNAs, mRNA-derived piRNAs, and transposon-derived piRNAs. Transposon-derived piRNAs are transcribed from two genomic strands, thus producing both piRNAs and antisense piRNAs; mRNA-derived piRNAs are usually derived from 3′ untranslated regions (UTRs), and lncRNA-derived piRNAs are produced from the entire transcript [35]. These precursors are usually generated by specific genomic locations containing repeating elements, and usually occur independently of the dicer. In addition, piRNAs require post-transcriptional modification to become mature piRNAs; piRNAs bear 2′-*O*-methyl-modified 3′ termini and guide PIWI-clade argonautes (PIWI proteins) rather than the AGO-clade proteins, which function in the miRNA and siRNA pathways [36,37,38,39,40,41,42,43] (Figure 1A).

PIWI proteins are germline-specific Ago family members [44] that are essential for germline development and gametogenesis in animals [39,45,46,47,48]. The PIWI protein family shares a conserved structure and function across multiple organisms [49], including fruit fly (PIWI, aubergine, and AGO3 proteins) [43], mouse (MILI, MIWI, and MIWI2) [39,50,51,52], human (HILI, HIWI1, HIWI2, and HIWIL3) [42,43,53,54], zebrafish (ZILI and ZIWI) [55], and nematode (PRG-1 and PRG-2) [56]. The classical function of PIWI/piRNAs is to maintain genomic integrity by repressing the mobilization of transposable elements, and to regulate the expression of downstream target genes via transcriptional or post-transcriptional mechanisms, including epigenetic silencing of transposons through DNA methylation [33,44] and H3K9 tri-methylation through recruitment of heterochromatin protein 1 (HP1) and histone methyltransferases (HMTs) [57]. piRNAs regulate mRNA levels by complementary sequence binding to the 3′UTR, and a protein’s stability by binding to it (Figure 1B). For instance, piRNA-54265 binds with the PIWIL2 protein and promotes the formation of the PIWIL2/STAT3/phosphorylated-SRC (p-SRC) complex, which activates STAT3 signaling and promotes the proliferation, metastasis, and chemo-resistance of colorectal cancer cells.

Recently the reactivation of PIWI protein expression, primarily PIWI-like proteins (PIWIL1, PIWIL2, PIWIL3, and PIWIL4), has been identified in various malignancies [58,59,60,61].

Increasing evidence suggests that PIWI proteins are linked to the hallmarks of cancer, such as cell proliferation, anti-apoptosis, genomic instability, invasion, and metastasis. Due to their restricted expression PIWI are classified as cancer/testis antigen (CTA) and are considered as excellent targets for diagnostic and prognostic biomarkers, and immunotherapy [62]. The expression of piRNA or PIWI protein in non-germline cancers is in line with the well-established phenomenon of cancer/germline genes, which describes the aberrant expression of germline-specific genes in non-germline cancers [61,63,64,65,66]. This provides new possibilities for anticancer therapies through the targeting of PIWI proteins, and which may have fewer side effects due to their restricted expression.

#### 2.1.1. piRNA in Gynecological Cancers

Increasing functional evidence supports the involvement of piRNAs in the regulation of epigenetic changes in tumorigenesis [67,68,69], along with posttranscriptional mRNA and protein stability regulation. It has been suggested that the PIWI–piRNA complex contributes to cancer development and progression by promoting a stem-like state of cancer cells, or cancer stem cells (CSCs). It has been reported that CSCs represent the cells that have undergone epithelial–mesenchymal transition (EMT) and acquired metastatic capacities. Such epigenetic alterations allow cancer cells to adapt to changes in their microenvironment. Epigenetic global changes in cancer include DNA hypo-methylation, histone hypo-acetylation, and gene-specific DNA hyper-methylation, leading to oncogene activation (R-ras, cyclin D2) [70], and tumor suppressor silencing (RB1, p16) [71]. In cancer tissues, aberrantly expressed piRNAs implicate global hypo-methylation and local hyper-methylation as potential cancer-specific features [69,72].

Abnormal expression of piRNAs is emerging as a crucial regulator in cancer cell proliferation, apoptosis, invasion, and migration. In gynecologic malignancies the study of piRNA expression and their pathophysiological significance remains exploratory. Singh et al. [73] used RNA sequencing to identify piRNAs in normal ovary (pi RNA #219), endometrioid (pi RNA #256), and serous ovarian cancer (pi RNA #234). Although sample numbers used in the study were small, the authors reported 159 and 143 differentially expressed piRNAs in endometrioid and serous ovarian cancer, respectively. The differentially expressed piRNAs in endometrioid ovarian cancer were comprised of 74 upregulated and 77 down-regulated piRNAs, while those in serous ovarian cancer included 56 upregulated and 81 downregulated. Specific findings showed that piR-52207 was upregulated in endometrioid ovarian cancer, and piR-52207 and piR-33733 were increased in serous ovarian cancer [73]. Upregulated piR-52207 targets NUDT4, MTR, EIF2S3, and MPHOSPH8, which promote endometrioid ovarian cancer cell proliferation, migration, and tumorigenesis. In serous ovarian cancer, piR-33733 targets LIAS3′-UTRs, whereas piR-52207 binds ACTR10 and PLEKHA5 3′-UTRs and 5′-UTRs, leading to increased anti-apoptotic and decreased pro-apoptotic proteins. Thus, piR-52207 and piR-33733 promote ovarian cancer oncogenes via involvement in multiple cell-signaling pathways at the post-transcriptional level, supporting them as possible therapeutic targets for ovarian cancer [73].

In endometrial cancer, studies utilizing small-RNA sequencing and microarrays, have shown a significant difference in the expression pattern of piRNAs between normal, hyperplastic, and the neoplastic endometrium [74]. Utilizing the stringent thermodynamic parameters for RNA–RNA binding, the authors showed each piRNA to be complementary to a number of mRNAs, ranging from 28 to 308. In total, 1526 mRNA targets were predicted; differential expression analysis on paired sample groups revealed that 170 of the predicted 1526 were differentially expressed (|FC| ≥ 1.5 and *p*-value 0.001) in hyperplastic and/or tumor tissues [74]. In cervical cancer cell lines piR-651 has been shown to be upregulated [75] however, apart from this one example the details of piRNA status in cervical cancer remains largely unexplored.

#### 2.1.2. Future Perspectives

Cancer and germ cells share several essential characteristics, including high proliferation rates and self-renewal abilities; in addition, cancer cells may re-activate cancer testis antigen (CTA) genes, whose expression is usually restricted to the germline, and silenced in adult somatic tissues. The expression of germline genes in cancer reflects the aberrant activation of a silenced developmental program that leads to escape from cell death, immune evasion, and invasiveness, thus contributing to the molecular mechanisms of carcinogenesis [61]. Among cancer testis antigens, PIWI-like (PIWIL) genes, belonging to the Ago family, are frequently deregulated in several malignancies, including cervical [60,76,77], endometrial [78], and ovarian cancer [58,59,76], and these proteins, along with piRNA, are involved in various aspects of malignancy, and associated with advanced tumor stage and poor prognosis. Along with piRNA, the PIWI proteins are prominently expressed in cancer cells, making them both useful biomarkers for cancer diagnosis and possible druggable targets [79].

### 2.2. tRNA-Derived Small RNAs

Transfer RNA (tRNA), one of the most abundant cellular ncRNAs, is important for protein translation. Recent research has shown that tRNAs are not always the terminally differentiated molecules; fragments derived from tRNAs are a source of small regulatory RNAs, known as tRNA-derived small RNAs (tsRNAs) [80]. Based on the cleavage site, tsRNAs can be divided into two main types: (1) transfer RNA-derived RNA fragments (tRFs) approximately 14 to 30 nucleotides in length, and derived from mature or precursor tRNAs; and (2) tRNA halves or tiRNAs, 29 to 50 nucleotides in length, induced by stress, and produced by specific cleavage at the anticodon loop of mature tRNA. tRFs are further subdivided into tRF-1s, tRF-3s, tRF-5s, and internal tRFs (i-tRFs or tRF-2s), while tiRNAs are divided into 5′tiRNA and 3′tiRNA (Figure 2) [81].

The tRF-1, also known as 3′U-tRF, originates from the 3′ untranslated regions (UTR) of pre-tRNA through RNase Z digestion, with the characteristic of a poly-U sequence [82]. The tRF-5s are generated from cleavage in the D-loop or the arm region between the D-loop and anticodon loop of mature tRNA, and include the intact sequence of the 5′ end of mature tRNA [83]. The tRF-3s originate from cleavage in the T-loop, and end with trinucleotides “CCA” [84]. The i-tRFs derive from an internal region of mature tRNA, and include the anticodon loop and part of the D- and T-loops [85] (Figure 2). Most human i-tRFs are 20 or 36 nucleotides long [86]. The tRF-5s occur predominantly in the nucleus; large numbers of tRF-5s are present in HeLa cell nucleoli [83]. tRF-3s and tRF-1 are more abundant in the whole cell fraction than the nuclear fraction, suggesting that both species occur exclusively in the cytoplasm [87]. Each class of tRFs is generated by specific ribonucleases and regulated by specific pathways.

One of the earliest discovered classes of tsRNAs were the stress (and starvation) induced tRNA fragments called tiRNA (tiR), or tRNA halves [88]. In mammals, tiRNAs are generated through cleavage by ribonuclease (RNAse (A or T)) angiogenin (ANG) within the anticodon loops of mature tRNAs [89]. This ANG cleavage produces two types of tiRNA: a tiRNA-5 and a tiRNA-3, which are the 5′ and 3′ half of mature tRNA, respectively [89] (Figure 2). The production of tiRNAs is induced by stress, such as starvation, oxidative stress, heat shock, UV irradiation, or viral infection [88,90]. The upregulation of ANG under certain conditions positively correlates with increased tiRNA levels [91,92,93]. RNH1, an ANG inhibitor interacting with ANG in the cytoplasm, is a negative regulator of tiRNA generation [90]. Several studies have shown that methylation of mature tRNA by DNA methyltransferase DNMT2 or cytosine-5 methyltransferase NSun2 enhances tRNA resistance to ANG cleavage [94,95].

Both tRFs and tiRNAs play important roles in tumorigenesis [85,86,87,96,97], and are promising diagnostic biomarkers and therapeutic targets for cancer. tRFs can modulate protein translation and interact with ribosomes and aminoacyl tRNA synthetases [98,99]. In addition, tRFs can associate with Ago and PIWI proteins in a cell-type specific manner, potentially affecting gene expression (Figure 3). Moreover, the interaction between RNA-binding proteins and tRFs has been linked with cancer development and metastasis [85,100].

In cancer models, tsRNAs promote cell proliferation and cell cycle progression by regulating the expression of oncogenes and proto-oncogenes. Functional studies show that tRFs and tiRNAs may bind to RNA binding proteins such as Y-box binding protein 1 (YBX1) and prevent transcription, inactivate initiation factor eIF4G/A, promote translation of ribosomal proteins, activate aurora kinase A (the regulator of mitosis) by binding to cytochrome C, or promote the assembly of stress granules that helps cells survive under adverse conditions [105,106,107,108]. A recent study indicated that two specific tRFs derived from tRNA^Lys-CTT^ and tRNA^Phe-GAA^ are good indicators of progression free survival (PFS), and thus are candidate prognostic markers in prostate cancer [106].

#### 2.2.1. tRNA Derived Small RNAs in Gynecologic Cancer

A significant impact of the deregulated tsRNAs has been demonstrated in various malignancies, including gynecological cancers; these biological functions of tRFs are Ago-dependent. In ovarian cancer tRF5^Glu^ regulates breast cancer anti-estrogen resistance 3 (BCAR3) mRNA levels by direct binding to the 3′ untranslated region (UTR) [109]. In colon, breast, and ovarian cancer patients, as well as corresponding cell lines, the expression level of ts-101 and ts-46 (tRF-1s) correlates with chromatin structure, cell survival, cell proliferation, clonal growth, and apoptosis. The expression of tRFs also correlates with oncogene activation and ovarian cancer progression [110]. Reanalysis of existing RNA-sequencing data, from 180 serum samples, including 15 healthy controls, 46 benign tumors, 22 borderline tumors, and 97 ovarian cancer patients, revealed that tsRNAs cover a high proportion of total small RNA, and are non-random degradation products in serum (ranging from 2.5–29.4%), and which are enriched in several specific types of related tRNA (e.g., Gly-tRNA), and can predict abnormal cell proliferation with high accuracy [111]. Another group using serum samples from ovarian cancer patients and healthy donors, along with ovarian cancer cell lines, have shown differential expression of tRF; they showed that tRF-03357 promoted SK-OV-3 cell proliferation, migration, and invasion, as well as downregulating HMBOX1 [112]. In cervical cancer, preliminary studies using biopsy samples demonstrated that the expression of 5S rRNA, tRNA^Arg^, and tRNA^Sec^ was significantly elevated in the HPV16-containing samples when compared to the HPV-negative biopsies [113]. In Hela cells, 5′ tRFs derived from tRNA^Gln^ are produced in abundance [83]. Currently, tRNA/tiRNA status in endometrial cancer remains unexplored.

#### 2.2.2. Future Perspectives

tsRNAs are unique sequences derived from tRNA precursors, and generated in the nucleus. tsRNAs are frequently dysregulated in various cancers, including the gynecologic malignancies. Since tsRNAs can accompany both Ago proteins (like miR) and PIWI proteins (like piRNAs), they can regulate gene expression both pre-transcriptionally (like piRNA) and post-transcriptionally (like miR). Like piRNAs, tsRNAs are produced as single-stranded molecules, and can interact with DNA and the histone methylation machinery, suggesting a role in the pre-transcriptional regulation of gene expression. Like miRNAs, ts-53 (previously known as miR-3676) interacts with the 3′UTR of TCL1, supporting a role for tsRNAs in the posttranscriptional regulation of gene expression [81,114].

Rapidly proliferating tumor cells often overcome a deficient blood supply, resulting in a microenvironment with limited oxygen and nutrients. Tumor cells adapt to this stress with varying strategies, thus ensuring survival and proliferation [115]. Generation of tsRNA from tRNAs under stress is an important pathway, and the biological function of tsRNA mainly supports cell survival under stress. Additionally, tsRNAs can be detected in the urine and serum from cancer patients [111,116,117,118], suggesting their potential as molecular diagnostic markers. For example, high-throughput RNA sequencing in breast cancer patients showed that tsRNA blood levels closely relate to the pathological characteristics [119]; similarly, in breast cancer and prostate cancer, hormone-dependent tsRNA are expressed in abundance and enhance the proliferation of cancer cells [91]. Thus, a tsRNA database of different tumors may represent a new diagnostic tool for cancer management [114,120]. In summary, tsRNAs are implicated in tumor onset, progression, and drug response, and thus represent potential therapeutic targets and/or diagnostic markers.

### 2.3. Micro-Ribonucleic Acid (miR)

Micro-ribonucleic acids (miRs/micro-RNAs) are short (18–25 nucleotides), evolutionarily conserved, and endogenously expressed regulatory RNA molecules, which belong to the family of ncRNAs. The miRs were first detected in the early 1990s in *Caenorhabditis elegans* [121], and later studies confirmed their presence in almost all species [122,123]. Although most miRs modestly alter expression of their target genes, the intricate network of miR target genes and downstream effectors plays a profound role in the regulation of biological pathways [124,125,126].

Among the non-coding RNAs, miR are the most widely studied, and apart from classical 3′ mRNA targeting and cytosolic expression, miR are also known for their nuclear expression [127]; the biogenesis and mRNA regulatory functions of miR have been extensively reviewed by Jacob O’Brien and others [30,31,128,129]. In brief, miRs are synthesized in the nucleus by DNA polymerase II [130] as a long double-stranded precursor called pri-miR. This pri-miR is cleaved at specific sites by the RNAse drosha inside the nucleus, producing a precursor miR (pre-miR) [131]. The pre-miR is exported to the cytoplasm by the exportin 5 protein, where it is processed by dicer into mature miR. Mature miRs are then activated through binding to the RNA-induced silencing complex (RISC); via the RISC, miRs can regulate their target mRNAs, leading to translational repression or degradation. The sequence at the 5′ end of the mature miR is called the “seed region”, or “seed sequence”. The seed sequence of the miR binds the complementary sequence within the 3′ untranslated region (3′ UTR) of target mRNAs [131,132]. Some miRs also interact with the 5′ UTR [133], coding sequence [134], and promoter regions of their targets [135]. Perfect or near-perfect complementarity between the miR and its mRNA target results in mRNA degradation, while imperfect complementarity leads to translational inhibition [136]. Several parameters, including the subcellular location of the miR, the quantity of both miR and its target mRNA, and the affinity, modulate the miR–mRNA interaction [128].

miRs are the critical modulators that regulate gene expression at the post-transcriptional level; they govern the stability and translation of protein coding mRNAs and are thus involved in almost every biological process, including regulation of cell division, differentiation, growth, and apoptosis. Dysregulation of miRs is instrumental in various pathophysiologies including neurodegenerative, inflammatory, metabolic, and cardiovascular diseases and, significantly for this review, cancer [137,138]. Altered expression of hundreds of miRs is closely associated with tumor development, invasion, metastasis, and drug resistance in cancer [139,140]. The miR expression profile differs significantly between normal and cancerous tissues, localized and aggressive cancer, and across type and stages of cancer.

Differential miR expression in normal and tumor cells has been reported in numerous studies, and suggests a robust association of altered miR expression with cancer pathogenesis and progression [141,142]. More than half of all annotated human miR genes are located in cancer-associated genomic regions that are amplified, deleted, or translocated in cancer [143]. Mechanistically, miRs regulate tumor suppressor genes and oncogenes; based on the regulated gene, miRs can be either oncogenic or tumor suppressive. The altered expression of miRs that control tumor suppressor genes and oncogenes results in cancer. Dysregulated miR expression has several causes, including genetic alteration, epigenetic changes, and SNPs in miR coding genes, as well as defects in factors regulating miR biogenesis.

#### 2.3.1. miR in Gynecologic Cancer

miRs are estimated to control over 50% of the activities of all protein coding genes [144], and are involved in regulation of almost all cellular processes [145]. Multiple studies have shown that dysregulation of miRs leads to a variety of human diseases, including cancer [146]. The relationship between miR expression and gynecologic cancer is well documented [147,148,149,150], but the significance of the cumulative effect of miR expression may not have been fully realized as each miR targets multiple genes associated with various cellular processes. Herein we summarize the key research findings on the significance of miR in gynecological cancer progression, prognosis, and diagnosis. We also present a compressive analysis of the miR expression in ovarian, endometrial, and cervical cancer, grouping them based on their expression. These miR are further analyzed using a systems biology approach for target prediction, and these target proteins are then analyzed for associated pathways to find the significance of altered miR in respective gynecological cancers. Based on the number of predicted target genes and associated pathways, miR can be best prioritized for possible disease interventions. A text-mining program called IRIDESCENT [151] was used to document details of the different miRs associated with ovarian, endometrial, and cervical cancer. IRIDESCENT identified the co-occurrences of miRNAs and gynecological cancer within MEDLINE titles and abstracts by analyzing public databases (e.g., Entrez, OMIM, and Disease Ontology) for a thesaurus of names and synonyms of miRNAs and diseases. The strength of association between gynecologic cancer and miR was calculated by the frequency of co-mention, (+0.5 for every abstract, +0.8 for every sentence). Associations and directionality, i.e., up- or downregulation, was confirmed manually for the respective types of cancer. Deregulated miRs were further grouped based on their expression in respective cancers, and potential significance was analyzed using miRNET [152].

##### Role of miRs in Ovarian Cancer Pathogenesis

Ovarian cancer is characterized by wide-scale deregulation of miRs, and aberrant expression of miRs in ovarian cancer is known to correlate with histotype, lymphovascular, and organ invasion [153]. We have reported that miR-15a and miR-16 are under-expressed in ovarian cancer tumor samples compared to normal tissue, and that ectopic expression of these miRs significantly inhibits ovarian cancer progression in preclinical tumor models [154,155]. Additionally, we have demonstrated that miR-195 is under-expressed in ovarian cancer and regulates ovarian cancer progression by regulating the expression of Mitochondrial Calcium Uptake 1 (MICU1) [156]. miR have been shown to play a key role in all the stages of tumorigenesis, and are also significant for diagnosis, prognosis, and therapeutics. Differential miR expression has been shown to be associated with ovarian cancer progression, omental metastasis, and drug resistance. Compared to normal ovarian tissue, miR-200a, miR-141, miR-200c, and miR-200b were significantly upregulated, whereas miR-199a, miR-140, miR-145, and miR-125b1 were among the most downregulated miRNAs in ovarian tumor tissues [153]. In another study miR-146a and miR-150 were reported to be significantly associated with omental metastasis and cisplatin resistance, and were shown to induce spheroid formation [157]. miR-200 and miR-429 are associated with recurrence and survival rates of ovarian cancer; their increased expression inhibits cancer metastasis [158].

The diagnosis of ovarian cancer has always been a significant problem; approximately 70–80% of patients present with advanced cancer at diagnosis. However, the results of miRNA research suggest that miR may be helpful in the early detection of ovarian cancer. Differences between the miR profiles of ovarian surface epithelium (OSE) and ovarian cancer, and the potential role of miRs in ovarian cancer diagnosis have been assessed in several studies [153,159,160,161,162,163,164,165,166,167]. In various reports miR-205, miR-429, miR-141, miR-200c, miR-93, miR-16, miR-20a, miR-21, miR-27a, miR-200a, miR-200b, and miR-200c [158,165,168] have been shown to be upregulated, while miR-320c, miR-383, let-7b, miR-99a, miR-125b, miR-145, miR-100, miR-31, miR-137, miR-132, and miR-26a were downregulated [163,165,167] in ovarian serous carcinoma samples. Other studies in ovarian cancer highlighted that miR levels can also discriminate between malignant and benign tumor, i.e., the expression pattern of let-7i-5p, miR-152, miR-122-5p, and miR-25-3p were significantly downregulated in malignant tumors compared to benign samples [160].

Presently, miRs are being evaluated for their potential utility as therapy candidates. In our previous reports, we have shown that the nanoliposomal delivery of miR-15a and miR-16 in combination, in a pre-clinical chemo-resistant orthotopic mouse model of ovarian cancer, demonstrated a striking reduction in tumor burden compared to cisplatin alone [154], while the importance of miR in cancer therapy has been reviewed previously [169,170].

Here, we have listed micro-RNAs deregulated in ovarian cancer using the text-mining program IRIDESCENT. The deregulated miRs were manually grouped by their expression levels compared to the control group (Table 1). Fifty-three miRs were reported to be upregulated, and sixty-eight miRs were downregulated in ovarian cancer. The miRNet web tool (https://www.mirnet.ca/ accessed between December 2020 and January 2021) was used for microRNA-gene target prediction (using miRTarBase v8.0 [171], and the parameters of degree and betweenness were used to derive the network, setting the cutoff for the degree filter at five. The network was further used to elucidate the biological processes and pathways of these upregulated, and downregulated targets, and networks are presented in Figure 4 and Figure 5. In the ovarian cancer upregulated miR group, a total of 7605 gene targets were found; of special note miR-20a-5p regulates 14.1% and miR 106a-5p regulates 9.4% of target genes. Kyoto Encyclopedia of Genes and Genomes (KEGG) pathway analysis enrichment was performed for these miR targets genes (degree cutoff 5); which revealed 24 significantly enriched pathways (cutoff *p* > 0.001), these altered pathways are known for their significant role in cancer progression, (Table 2). Gene ontology enrichment for biological process (GO-BP) and molecular function (GO-MF) were also analyzed; using *p* > 0.001 as the cutoff, GO-BP and GO-MF respectively identified 67 and 24 pathways as enriched, and the top 10 pathways are shown in Table 2.

Similar analyses for the downregulated miRs in ovarian cancer (Figure 2) identified 9287 gene targets. miR-26b-5p, miR-519d, miR-15a, and miR-15b regulated 20.8%, 11.3%, 8.6%, and 8.9% of the target genes, respectively. KEGG, GO-BP, and GO-MF enrichment analysis of the targets respectively identified 41, 95, and 38 enriched pathways. The top 10 pathways for each analysis are shown in Table 2.

While mir-21 and mir-155 are the most studied miRs in ovarian cancer, our analysis suggests that the genes targeted by miR-20a-5p and miR-26b-5p may play a major role in the progression of ovarian cancer (Figure 2). The most prominent targets of these miRs are phosphatase and tensin homolog (PTEN), cyclin dependent kinase inhibitor 1A (CDKN1A), MDM2 proto-oncogene (MDM2), superoxide dismutase 2 (SOD2), high mobility group box 1 (HMGB1), insulin like growth factor 1 receptor (IGF1R), WEE1 G2 checkpoint kinase (WEE1), forkhead box K1 (FOXK1), and thioredoxin interacting protein (TXNIP), which play a crucial role in ovarian cancer progression. From the pathway enrichment analysis, these deregulated miRs were shown to be involved in essential cellular processes and key signaling pathways, i.e., cell proliferation, regulation of cell cycle and apoptosis, adherens junction, focal adhesion, regulation of actin cytoskeleton, regulation of transcription, p53 signaling pathway, TGF-beta signaling pathway, Erbb signaling pathway, neurotrophin signaling pathway, Wnt signaling pathway, Jak-STAT signaling pathway, and insulin signaling pathways (Table 2). Our systemic analysis of these deregulated miRs, their target genes, and the target pathways, which are instrumental in cancer progression, reinforced the importance of these miRs in ovarian cancer, and may help prioritize miRs as targets for therapeutic evaluation.

##### Role of miRs in Endometrial Cancer Pathogenesis

The association of miR expression with the prognosis of endometrial cancer including lymph node metastasis, lymphovascular space invasion, overall survival, and recurrence-free survival is well documented [299,300,301,302,303,304]. Srivastava et al. [305] reviewed miRs differentially expressed between endometrial cancer and normal endometrial tissue, including upregulation of miR-9, miR-92a, miR-141, miR-182, miR-183, miR-186, miR-200a, miR-205a, miR-222, miR-223, miR-410, miR-429, miR-449, and miR-1228, and the downregulation of miR-99b, miR-143, miR-145, miR-193b, and miR-204. Numerous miRs regulate endometrial cancer cell proliferation by silencing their target genes [306,307], and miR expression profiles associate with stage, grade, relapse, and nodal metastases in endometrial cancer [301,305,308]; their importance has been reviewed [148].

Here, we have compiled a list of micro-RNAs that are deregulated in endometrial cancer using the text-mining program IRIDESCENT (Table 3). We report nineteen upregulated and twenty-seven downregulated miRs in endometrial cancer. The miRNet web tool (https://www.mirnet.ca/) was used for microRNA-gene target prediction, and the parameters of degree and betweenness were used to draw the network, and were further used to elucidate the predicted biological processes and pathways of these upregulated, and downregulated targets; the networks are presented in Figure 6 and Figure 7. In endometrial cancer, upregulated miR were predicted to target a total of 4224 genes. Among these target genes, 14.5% and ~26% are targeted by miR-21-5p and miR-106a-5p, respectively. Pathway enrichment analysis was performed on the target genes using KEGG, GO-BP, and GO-MF, as described above, and these methods respectively identified 32, 96, and 34 enriched pathways. A similar analysis was performed using the downregulated miRs; 5749 gene targets were identified, a network was generated with degree filter 5, and miR-195-5p and miR-424-3p respectively regulated 11.1% and 8.2% of the target genes. KEGG, GO-BP, and GO-MF enrichment analysis respectively identified 57, 100, and 97 pathways. The top 10 KEGG pathways, biological processes, and the molecular functions altered by the miRs are shown in Table 4. A literature search showed miR-205 to be the most studied miR in endometrial cancer, however, our target analysis showed miR-21-5p and miR-106-5p to be the most interconnected miRs that can regulate multiple targets responsible for endometrial cancer progression. The most prominent targets of these miRs are PTEN, IGF1R, Notch Receptor 2 (NOTCH2), cyclin D1 (CCND1), TXNIP, AGO2, and cyclin dependent kinase 6 (CDK6). The pathway enrichment analysis showed that these deregulated miRs are involved in critical cellular processes and key signaling pathways, i.e., cell cycle, proliferation, programmed cell death, cellular response to stress, p53 signaling pathway, mTOR signaling pathway, insulin signaling pathway, notch signaling pathway, NOD-like receptor signaling pathway, inositol phosphate metabolism, and adipocytokine signaling pathway (Table 4), all of which support the importance of these miRs in the progression of endometrial cancer.

##### Role of miRs in Cervical Cancer Pathogenesis

miR deregulation plays a key role in the malignant transformation of cervical cancer, and, along with their target genes, miRs have been exploited for both prognostic and therapeutic strategies in cervical cancer. HPV causes about 91% of cervical cancers [354], and the high-risk HPV strains HPV-16 and HPV-18 in keratinocyte cell lines have been shown to regulate expression of different miRs [237]. The study reports that both HPV strains induced the expression of miR-16, miR-25, miR-92a, and miR-378, while they inhibited the expression of miR-22, miR-27a, miR-29a, and miR-100. Moreover, data from cervical-tissue specimens correlated with in vitro results; miR-25, miR-92a, and miR-378 were increased with cervical cancer progression, which supports the significance of HPV mediated miR expression in cervical cancer pathogenesis [237].

Mounting evidence supports the significance of miR expression in cervical cancer, e.g., miR-21, which is reportedly overexpressed in various malignancies [355], promotes cell proliferation in HeLa cervical carcinoma cells by targeting programmed cell death 4 (PDCD4) expression. In addition, miR-20a [356], as well as miR-106a [357], suppressed the migration and invasion of cervical cancer cells by targeting TIMP2.

The potential of miR for early detection and as a prognostic biomarker in cervical cancer has also been explored. Expression profiles of circulating miRs in cervical cancer patient samples compared to healthy volunteers demonstrated that the expression of miR-20a and miR-203 was upregulated in the cervical cancer patients’ sera, and circulating levels of miR-20a could be useful for the detection of lymph-node metastasis [358]. Various reports have shown the correlation of miR expression with cervical cancer stage; miR-494 is upregulated, while the expression of miR-195 and miR-144 was downregulated in patients with stage IB cervical carcinoma [359,360,361]. In Stages I and II, the levels of miR-375, miR-145, and miR-124 were downregulated, while miR-99a/b, miR-92a, miR-150, and miR-21 levels were upregulated [359,362,363,364,365,366,367,368]. In cervical cancer patients, levels of miR-218 were similarly decreased in plasma and tumor samples, and levels correlated with tumor stage [369]. Differentially expressed miRs in the serum of cervical cancer patients, compared with controls, measured using Solexa sequencing, identified 12 miRs of interest. Five were upregulated, i.e., miR-21, miR-29a, miR-200a, miR-25, and miR-485-5p, and likely are credible alternatives to current tumor markers, such as squamous cell carcinoma (SCC) antigen and carbohydrate antigen 125 (CA125) [370]. Elevated serum levels of miR-205 correlate with both cervical cancer tumor stage and decreased survival rates. Some of the aforementioned miRs may be viable diagnostic and prognostic biomarkers for cervical cancer progression. Circulating levels of miR-646, miR-141, and miR-542-3p are significantly different when assayed before and after surgical procedures [371], and may be useful in monitoring the health status of patients after treatment.

The role of miRs in cervical cancer progression [372,373,374,375,376], therapeutics [150,377], and drug resistance is well documented [378]. The association of miRs with HPV infection is also well established, and includes cellular miR-9 [379], cellular miR-21 [41,51,148], exosomal vesicles (EV)-derived miR-21 [380,381], cellular miR-34a [376,381], EV-derived miR-34a [381], cellular miR-146a [382], and EV-derived miR-146a [380]. 

Here, we have compiled a list of micro-RNAs deregulated in cervical cancer (Table 5); thirty-six miRs were reported to be upregulated and seventy downregulated in cervical cancer. The miRNet web tool, as discussed above, was used for target gene prediction, and parameters of degree and betweenness were used to draw the network, and were further used to elucidate the predicted biological processes and pathways of the upregulated, and downregulated targets. The miR and the target protein networks are shown in Figure 8 and Figure 9. In cervical cancer, target prediction using upregulated miR resulted in 6957 gene targets; mir-106b-and miR-106a respectively regulate 15.7% and 10.3% of the target genes. Networks for these targets were generated and pathway enrichment analysis performed using KEGG, GO-BP, and GO-MF parameters, as before; they identified 19, 94, and 10 enriched pathways, respectively (Table 6). A similar analysis with the downregulated miR, generated 9715 gene targets, of which miR15a-5p targeted 7.4%, and miR195-5p regulated 6.6%. KEGG, GO-BP, and GO-MF enrichment analyses were applied and respectively identified 51, 96, and 55 pathways (Table 6). From our collective analysis of miR and their target genes in cervical cancer based on the number of target genes and associated networks, the most important upregulated miRs were miR-106, miR-20a, and miR-519d, while the most important downregulated were miR-15a and miR-195. The most prominent targets of these miRs are PTEN, CRK proto-oncogene, adaptor protein (CRK), SOD2, TXNIP IGF1R, BCL2 apoptosis regulator (BCL2), and FOXK1, which play a crucial role in cervical cancer progression. Based on the network analysis these deregulated miRs are involved in important cellular pathways and key signaling processes, i.e., cell cycle, adherens junction, pyrimidine metabolism, focal adhesion, cell proliferation, nuclear transport; and the p53 signaling pathway, Wnt signaling pathway, PPAR signaling pathway, G1 phase of mitotic cell cycle, ErbB signaling pathway, insulin signaling pathway, and MAPK signaling pathway (Table 6). All these cellular processes and signaling pathways are instrumental in the progression of cervical cancer, and further substantiate the significance of these miRs in cervical cancer.

#### 2.3.2. Future Perspectives

Ongoing research reveals the diagnostic potential and therapeutic promise of miRs for multiple cancers, including those of the female reproductive system. However, their diagnostic and/or therapeutic translation in terms of helping patients in need has not been fully apprehended, and before this becomes a reality, our understanding of the diverse roles of miRs must become complete. There is a certain level of redundancy in gene regulation by miRs, as a single miR can target multiple genes, and a single gene can be the target for multiple miRs. Thus, any miR selected as a therapeutic target must be comprehensively investigated, all target genes must be identified, and the outcome(s) of their combined inhibition should be carefully evaluated. Conflicting reports on the functionality of a single miR within a specific gynecologic cancer, as well as across different gynecologic cancers, suggest that we do not fully understand the complex nature of miR-mediated regulation, which may be tissue and cell-context specific. As an example, miR-26a was first shown to promote ovarian cancer proliferation through its suppression of ER-α [481], while subsequently, it was shown to inhibit proliferation of ovarian cancer cells through its regulation of CDC6 [482]. In a cervical cancer model, miR-26a inhibited cell proliferation, migration, and invasion in vitro, and also inhibited tumor growth in vivo in a xenograft model [483]. Similarly, miR-31 [484,485,486], miR-214 [181,487], miR-494 [361,488,489], and miR-222 [490,491] have different reported roles in different gynecologic cancers. This conflict arises from the targeting of multiple genes by the same miR, and the effects must be evaluated at a systems level using systems biology approaches in order to translate miRs into successful cancer therapies. Enrichment analysis based on the validated/predicted targets, as we have shown herein, will undoubtedly help in selecting the best miRs for a specific diagnostic, therapeutic, or predictive purpose. Concurrent with efforts to overcome the inherent barriers to miR use, strategies to manipulate miR levels in cancer patients should be developed further. This will permit the rapid translation of constructive findings to the clinic for the benefit of millions of gynecologic cancer patients worldwide.

## 3. Conclusions

We remain in the early stages of decoding the essential role of the small noncoding RNAs which were previously believed to be transcriptional debris. It is now well established that these small non-coding RNAs play a key role in gene regulation, and have a profound role in the development of multiple cancers, including gynecologic malignancies. Our understanding of the roles of piRNA- and tRNA-derived small RNAs is still very limited; interestingly, much valuable information is available for miR, and a few studies are at the preliminary stages of development for cancer therapy, with some clinical trials in progress for ovarian cancer (ClinicalTrials.gov Identifier: NCT03738319, NCT03776630, NCT01970696, NCT02253251, NCT01391351, NCT03877796), endometrial cancer (ClinicalTrials.gov Identifier: NCT02983279, NCT01119573, NCT03824613), and cervical cancer (ClinicalTrials.gov Identifier: NCT04087785). Based on this accumulating evidence, it is possible that in the near future miRs in plasma may be used for the diagnosis and prognosis of various cancers. One limitation of these miR expression data may be the lack of evaluation among diverse ethnic groups; however, as more information about the diverse types of small non-coding RNA and their target networks become available, it will assist us in refining their possible diagnostic and therapeutic applications. The primary causes of greater cancer related mortality, such as in ovarian cancer, are late diagnosis and acquired resistance; advancing the use of small noncoding RNA in cancer diagnosis and therapy will enhance early detection and therapeutic intervention, and will certainly be a significant step forward in the management of gynecologic malignancies.

## Figures and Tables

**Figure 1 cancers-13-01085-f001:**
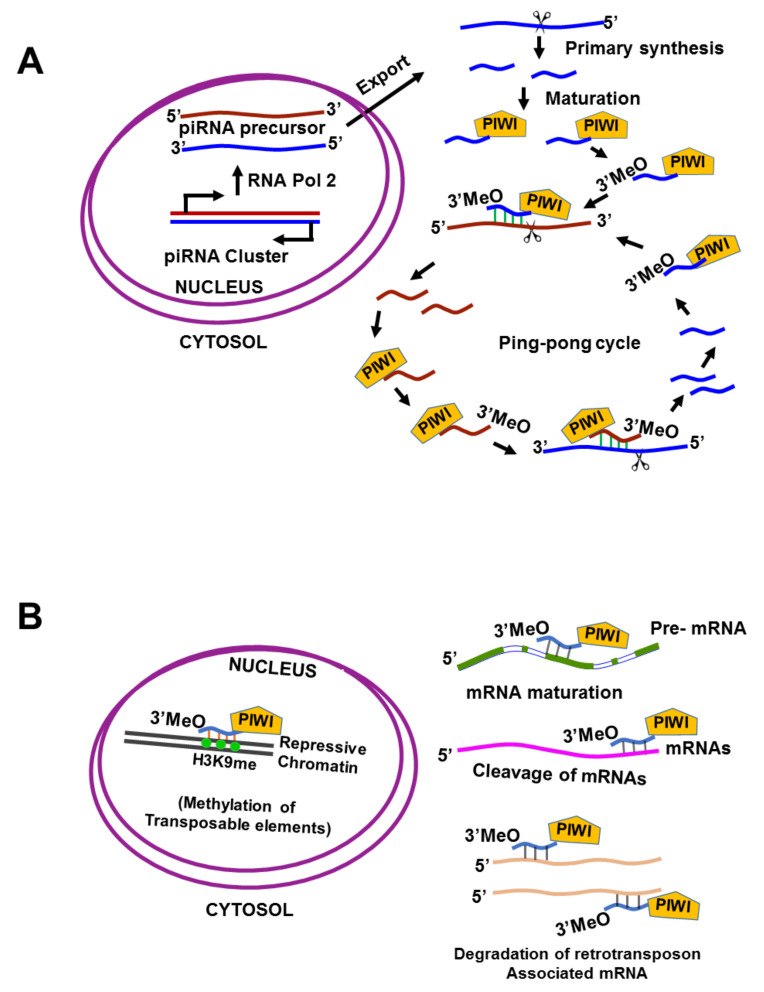
Biogenesis and function of piRNAs. (**A**) Biogenesis of piRNAs: Biogenesis of piRNA occurs through primary and secondary pathways (ping-pong cycle). In the primary pathway, piRNA precursors are transcribed from piRNA clusters by RNA polymerase 2 (RNA Pol2). Antisense primary piRNAs are cleaved, trimmed into short fragments, and their 3′ ends are 2′-*O*-methylated and then loaded onto PIWI family proteins. In the secondary amplification pathways, also known as the ping-pong cycle, PIWI proteins associate with antisense piRNA and cleave piRNA precursors in the sense strand, or PIWI proteins associate with sense piRNA and cleave antisense piRNA precursors in the sense strand. The incorporated RNA is therefore processed into a mature secondary piRNA by trimming and modification, likely by the same mechanisms that generate a primary piRNA. (**B**) The biological functions of piRNAs: In the nucleus, PIWI–piRNA complexes can repress the transposon expression by methylation of the transposon region or chromatin modification around the transposon region. In the cytoplasm, piRNAs can facilitate mRNA maturation, cause cleavage of mRNAs through a miRNA-like mechanism, and degrade retrotransposon-associated mRNAs.

**Figure 2 cancers-13-01085-f002:**
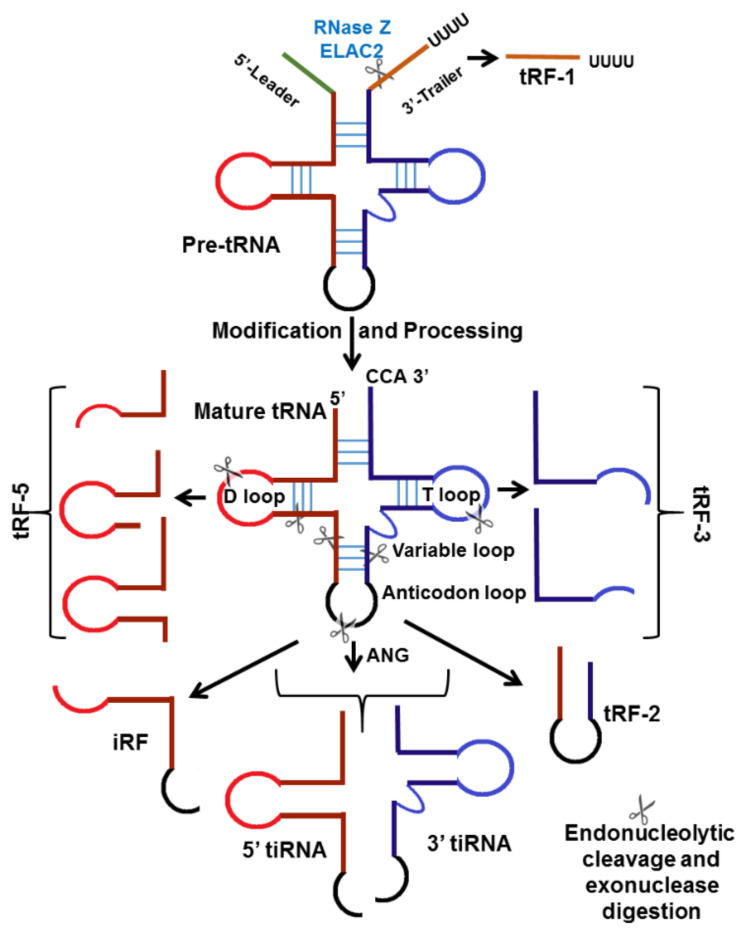
Biogenesis and function of tRNA-derived small RNAs: Different types of tRNA-derived RNA fragments produced from either pre-tRNAs or mature tRNAs. The tRF-1 series is produced by RNase Z (or ELAC2) cleavage of the pre-tRNA during tRNA processing. Mature tRNA can be cleaved in the anticodon loop by angiogenin (ANG) to produce the 5′-tiRNA and 3′-tiRNA series under stress conditions. tRF-2 is a tRNA fragment containing an anti-codon loop generated by an unknown cleavage method. Cleavage in the T-loop results in the production of the 3′-tRF series. The 5′-tRF series is derived from the 5′-end of mature tRNAs by endonucleolytic cleavage and exonuclease digestion in the D-loop.

**Figure 3 cancers-13-01085-f003:**
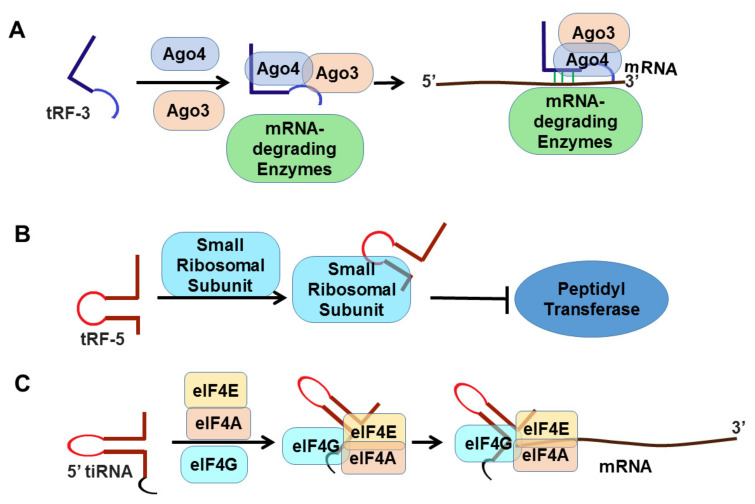
tRF- and tiRNA-mediated gene regulation: (**A**) Regulation of mRNA stability: The combination of tRF-3 with argonaute 3 (Ago3) and Ago4 binding to mRNA allows mRNA-degrading enzymes to degrade the target mRNA [101]; tRFs function like miRNAs to inhibit cancer-associated gene expression. Binding with the 3′ untranslated region (3′UTR) of target mRNA, Argonaute (Ago) protein and other proteins form an RNA-induced silencing complex (RISC). (**B**,**C**) Inhibition of translation: By binding to small ribosomal subunits, tRF inhibits peptidyl transferase activity that results in reduced protein abundance of the target gene [99,102]. 5′tiRNA inhibits translation by forming a RNA G-quadruplex (RG4s) that replaces the translation initiation complex eIF4G/eIF4E on the mRNA cap [103,104].

**Figure 4 cancers-13-01085-f004:**
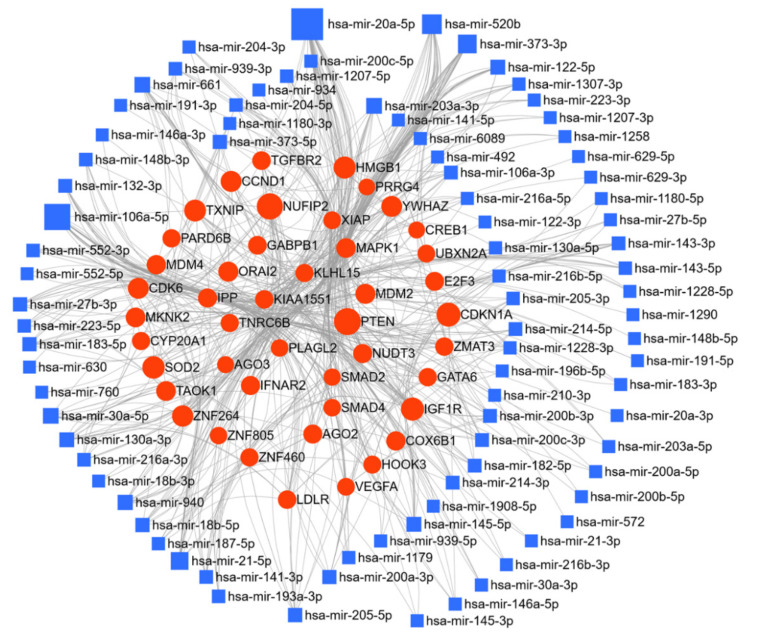
Network of upregulated miR in ovarian cancer: miR deregulated in ovarian cancer were searched for in the PubMed database using IRIDESCENT. The network of upregulated miR generated using miRNET (degree filter cutoff 10) is shown. The blue squares represent miR, and the red dots represent target genes of these miRs.

**Figure 5 cancers-13-01085-f005:**
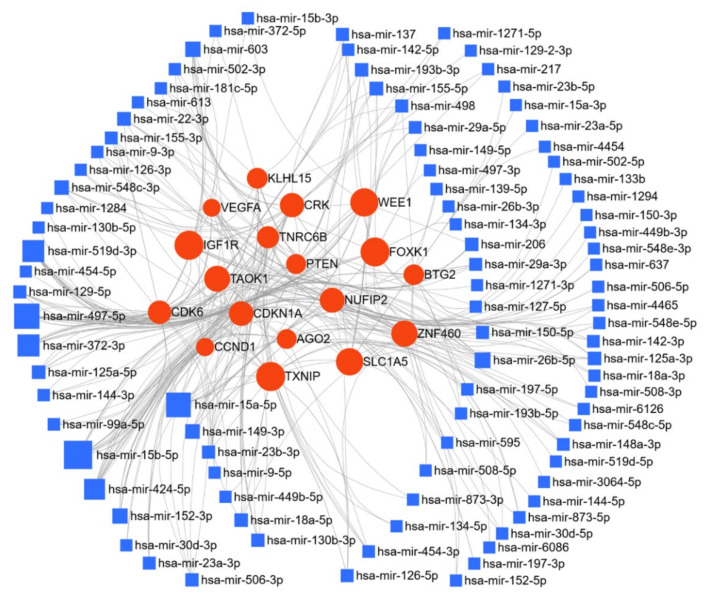
Network of downregulated miR in ovarian cancer: miR deregulated in ovarian cancer were searched for in the PubMed database using IRIDESCENT. The network of downregulated miR generated using miRNET (degree filter cutoff 15) is shown. The blue squares represent miR, and the red dots represent target genes of these miRs.

**Figure 6 cancers-13-01085-f006:**
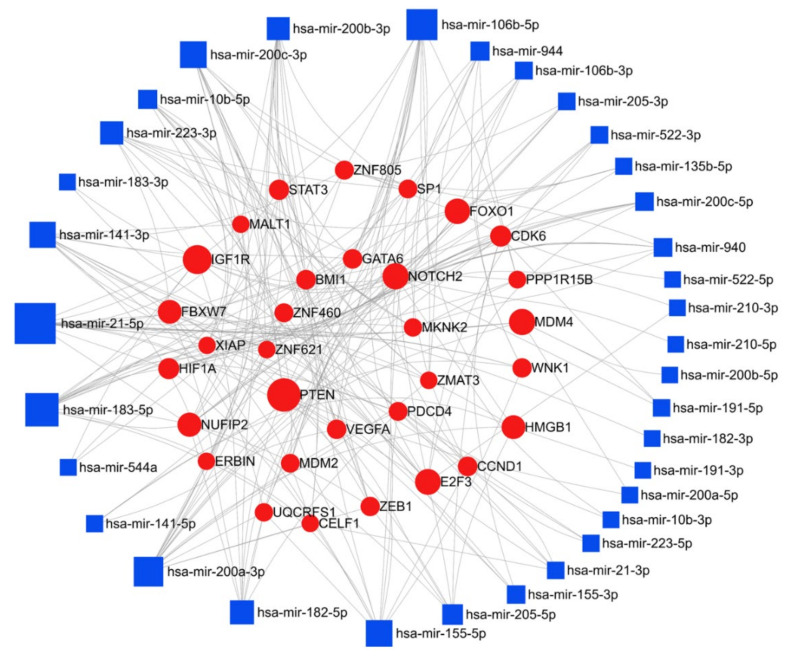
Network of upregulated miR in endometrial cancer: miR deregulated in endometrial cancer were searched in the PubMed database using IRIDESCENT. The network of upregulated miR generated using miRNET (degree filter 5) is shown. The blue squares represent miR, and the red dots represent target genes of these miRs.

**Figure 7 cancers-13-01085-f007:**
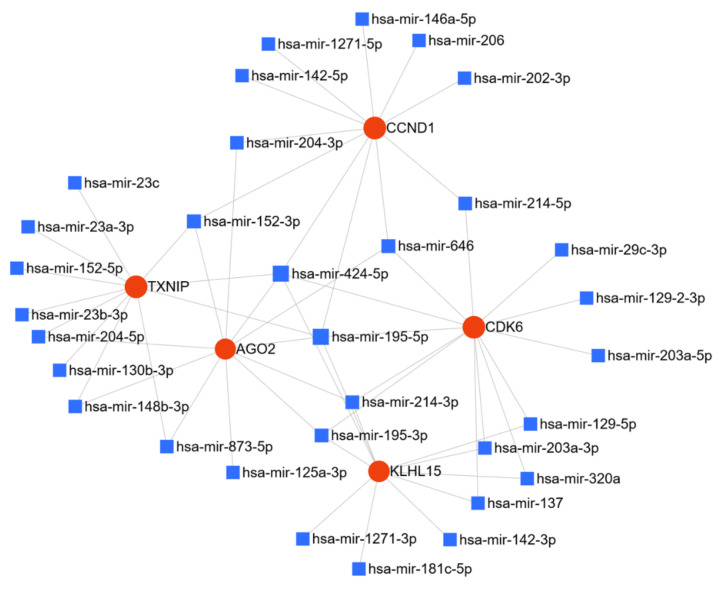
Network of downregulated miR in endometrial cancer: miR deregulated in endometrial cancer were searched in the PubMed database using IRIDESCENT. The network of downregulated miR generated using miRNET (degree filter cutoff 10) is shown. The blue squares represent miR, and the red dots represent target genes of these miRs.

**Figure 8 cancers-13-01085-f008:**
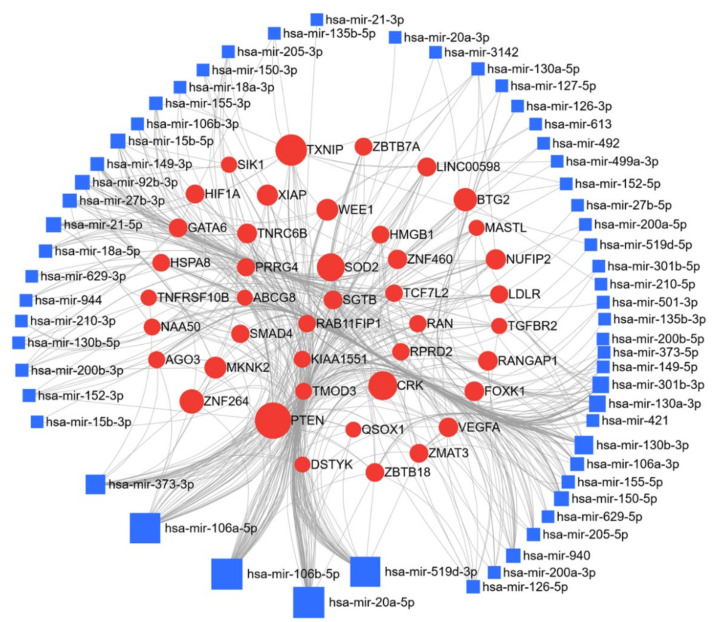
Network of upregulated miR in cervical cancer: miRs deregulated in cervical cancer were searched in the PubMed database using IRIDESCENT. The network of upregulated miRs generated using miRNET (degree filter cutoff 10) is shown. The blue squares represent miR, and the red dots represent target genes of these miRs.

**Figure 9 cancers-13-01085-f009:**
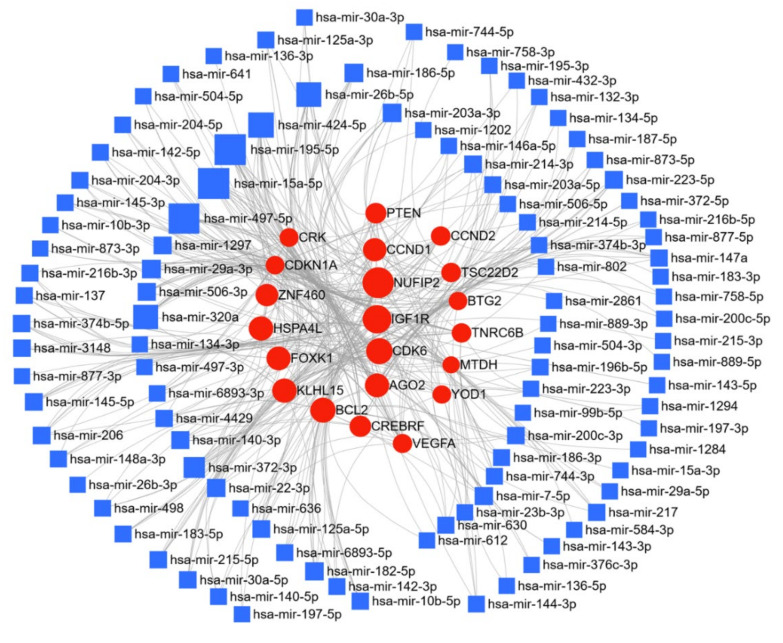
Network of downregulated miR in cervical cancer: miR deregulated in cervical cancer were searched in the PubMed database using IRIDESCENT. The network of downregulated miR generated using miRNET (degree filter 10) is shown. The blue squares represent miR, and the red dots represent target genes of these miRs.

**Table 1 cancers-13-01085-t001:** Deregulated miR in Ovarian Cancer.

Related Entity	Lit Str.	Lit MIM	Regulation	No. of Papers	Ref
miR-21	84.4	−1.9	Up	66	[172,173,174,175,176]
miR-200C	69.5	−0.1	Up	57	[177,178]
miR-145	75.2	−1	Up	48	[153,179,180]
miR-200A	52	0.2	Up	43	[181]
miR-200B	28.4	−0.1	Up	35	[174,179,182]
miR-141	30.1	−0.3	Up	33	[178,179,183]
miR-214	42.4	−0.6	Up	31	[181]
miR-205	52.3	−0.5	Up	30	[178,179,180,184]
miR-182	35.1	−0.6	Up	24	[184,185]
miR-146A	20.8	−2	Up	21	[186]
miR-210	13.5	−1.6	Up	15	[187]
miR-130A	23.6	−0.4	Up	14	[188]
miR-143	3.9	−2.8	Up	12	[163]
miR-30A	4.9	−1.6	Up	11	[189]
miR-106A	20.3	−0.9	Up	10	[190]
miR-204	4.9	−1.8	Up	10	[191]
miR-132	18.9	−1.5	Up	9	[163,192,193]
miR-20A	7.6	−2	Up	8	[194]
miR-183	5.2	−1.8	Up	7	[195]
miR-630	10.2	0.1	Up	6	[196]
miR-223	9.6	−1.9	Up	6	[197]
miR-216A	8.7	−1	Up	6	[198]
miR-373	4.4	−0.9	Up	6	[199]
miR-203A	39	−1.2	Up	5	[179,200]
miR-193A	7.6	−0.5	Up	5	[201]
miR-191	4.1	−0.7	Up	5	[202]
miR-1307	18	1.3	Up	4	[203]
miR-27B	9.2	−1.6	Up	4	[204]
miR-187	6	−0.3	Up	4	[205]
miR-492	8.7	0.1	Up	3	[206]
miR-760	8.2	−0.8	Up	3	[207]
miR-661	7.9	0.3	Up	3	[208]
miR-1181	7.1	1.2	Up	3	[209]
miR-1258	7.1	0.2	Up	3	[210]
miR-940	6.6	−1	Up	3	[211]
miR-1290	3.9	−0.7	Up	3	[212]
miR-148B	3.7	−2.2	Up	3	[213]
miR-939	3.7	−1	Up	3	[214]
miR-572	8.2	0.2	Up	2	[215]
miR-1180	7.4	0.6	Up	2	[216]
miR-18B	7.4	−0.6	Up	2	[217]
miR-1908	5.8	0	Up	2	[218]
miR-1228	4.5	0.1	Up	2	[219]
miR-122	4.2	−3.7	Up	2	[220]
miR-196B	4.2	−1.4	Up	2	[221]
miR-552	7.7	−0.7	Up	1	[222]
miR-1207	5.3	1.1	Up	1	[223]
miR-629	5.3	−0.6	Up	1	[224]
miR-520B	4.5	−1.3	Up	1	[225]
miR-934	4.2	1	Up	1	[226]
miR-1179	3.7	−0.5	Up	1	[227]
miR-216B	3.7	−1.9	Up	1	[228]
miR-6089	3.7	0.7	Up	1	[229]
miR-155	8.8	−3.1	Down	21	[230]
miR-137	38.5	−1	Down	16	[231,232,233]
miR-22	27.1	−1.1	Down	15	[234]
miR-506	15.9	0.1	Down	15	[235]
miR-23A	23.7	−1.2	Down	12	[236]
miR-206	21.1	−1.9	Down	12	[237]
miR-152	20	−0.7	Down	11	[238]
miR-30D	10.7	−0.2	Down	11	[239]
miR-148A	39.1	−0.8	Down	11	[240]
miR-130B	18.7	−0.7	Down	10	[241]
miR-126	11.3	−2.6	Down	10	[242]
miR-193B	10.8	−0.9	Down	9	[243]
miR-18A	20.9	−1.4	Down	8	[244]
miR-99A	10.8	−1.2	Down	8	[245]
miR-497	24.6	−1.1	Down	7	[246]
miR-23B	15.5	−1	Down	7	[247]
miR-29A	9	−2	Down	7	[248]
miR-125A	12.7	−1.8	Down	6	[249]
miR-134	11.3	−1.1	Down	6	[250]
miR-133B	23.3	−1.5	Down	5	[251]
miR-149	16.2	−2	Down	5	[252]
miR-26B	11.9	−1.8	Down	5	[253]
miR-150	7.3	−2	Down	5	[254,255]
miR-217	3.9	−1.7	Down	5	[256]
miR-144	17.7	−1.5	Down	4	[257]
miR-1271	16.7	−0.6	Down	4	[258]
miR-613	13.5	−1	Down	4	[259]
miR-498	12.4	−0.2	Down	4	[260]
miR-595	5.3	−0.3	Down	4	[261]
miR-424	4.7	−1.5	Down	4	[262]
miR-15A	3.9	−2.3	Down	4	[155]
miR-449B	3.9	0.5	Down	4	[263]
miR-139	12.2	−1.3	Down	3	[264]
miR-454	11.9	−0.4	Down	3	[152]
miR-15B	9	−2.1	Down	3	[265]
miR-181C	8.2	−1.6	Down	3	[266]
miR-136	7.1	−0.7	Down	3	[267]
miR-372	5.5	−0.7	Down	3	[268]
miR-1182	7.4	1.1	Down	2	[269]
miR-3064	7.4	1.6	Down	2	[270]
miR-519D	7.4	−0.6	Down	2	[271]
miR-LET7C	6.9	−2.6	Down	2	[272]
miR-508	6.6	1.1	Down	2	[273]
miR-598	5.8	0.1	Down	2	[274]
miR-6126	5.8	1.9	Down	2	[275]
miR-1294	5.3	−0.9	Down	2	[276]
miR-197	5.3	−1.7	Down	2	[277]
miR-1284	5	0.3	Down	2	[278]
miR-127	3.9	−0.8	Down	2	[279]
miR-637	3.7	−1	Down	2	[280]
miR-LET7B	10.6	−0.7	Down	1	[281]
miR-8073	9	1.6	Down	1	[282]
miR-LET7E	8.6	−0.1	Down	1	[283]
miR-LET7D	7.6	−0.6	Down	1	[284]
miR-9-1	6.3	0.5	Down	1	[285]
miR-603	6.1	−0.3	Down	1	[286]
miR-873	6.1	−1.3	Down	1	[287]
miR-LET7I	6	−0.5	Down	1	[189]
miR-548C	5.3	1.5	Down	1	[288]
miR-502	5	0.4	Down	1	[289,290]
miR-129-2	4.7	0.1	Down	1	[291]
miR-4454	4.5	0.5	Down	1	[292]
miR-548E	4.5	1.6	Down	1	[293]
miR-551A	4.5	0.5	Down	1	[294]
miR-4465	3.7	0.9	Down	1	[295]
miR-6086	3.7	1.2	Down	1	[296]
miR-634	3.7	−0.6	Down	1	[297]
miR-142	3.4	−1.5	Down	1	[298]

Deregulated miR in ovarian cancer. Lit Str. and Lit MIM represent: literature strength, and literature mutual information, respectively. The miR were regrouped based on their expression. The number of publications with the miR was retrieved from PubMed, and representative manuscripts are cited.

**Table 2 cancers-13-01085-t002:** Altered Pathways by Deregulated miR in Ovarian Cancer.

A. KEGG Pathways Regulated by Rpregulated miR				
S.No.	Name	Hits	*p*-Value	Adj. *p*-Value
1	Pathways in cancer	36	1.76 × 10^−12^	1.76 × 10^−10^
2	Chronic myeloid leukemia	18	3.93 × 10^−12^	1.97 × 10^−10^
3	Prostate cancer	18	9.44 × 10^−11^	3.15 × 10^−09^
4	p53 signaling pathway	16	1.44 × 10^−10^	3.60 × 10^−09^
5	Bladder cancer	11	4.30 × 10^−10^	8.60 × 10^−09^
6	Glioma	14	7.53 × 10^−09^	1.26 × 10^−07^
7	Pancreatic cancer	14	1.71 × 10^−08^	2.44 × 10^−07^
8	Cell cycle	18	3.73 × 10^−08^	4.66 × 10^−07^
9	Adherens junction	13	1.73 × 10^−07^	1.92 × 10^−06^
10	Melanoma	12	9.43 × 10^−07^	9.43 × 10^−06^
**B. Gene Ontology Enrichment for Biological Process (GO-BP) Regulated by Upregulated miR**				
**S.No.**	**Name**	**Hits**	***p*-Value**	**Adj. *p*-Value**
1	G1 phase of mitotic cell cycle	11	7.03 × 10^−08^	5.6 × 10^−06^
2	G1 phase	11	1.12 × 10^−07^	5.6 × 10^−06^
3	Gland development	27	2.20 × 10^−07^	5.98 × 10^−06^
4	Negative regulation of transcription from RNA polymerase II promoter	39	2.39 × 10^−07^	5.98 × 10^−06^
5	Regulation of cell proliferation	72	1.72 × 10^−06^	3.44 × 10^−05^
6	Negative regulation of transcription, DNA-dependent	54	4.07 × 10^−06^	4.67 × 10^−05^
7	Response to ionizing radiation	54	4.07 × 10^−06^	4.67 × 10^−05^
8	Negative regulation of cellular biosynthetic process	14	4.16 × 10^−06^	4.67 × 10^−05^
9	Negative regulation of RNA metabolic process	63	4.2 × 10^−06^	4.67 × 10^−05^
10	Negative regulation of nucleobase-containing compound metabolic process	55	5.68 × 10^−06^	5.45 × 10^−05^
**C. Gene Ontology Enrichment for Molecular Function (GO-MF) Regulated by Upregulated miR**				
**S. No.**	**Name**	**Hits**	***p*-Value**	**Adj. *p*-Value**
1	Negative regulation of transcription, DNA-dependent	54	1.02 × 10^−06^	0.000102
2	Double-stranded DNA binding	16	3.64 × 10^−06^	0.000182
3	Structure-specific DNA binding	20	1.31 × 10^−05^	0.00033
4	Sequence-specific DNA binding	41	1.32 × 10^−05^	0.00033
5	RNA polymerase II distal enhancer sequence-specific DNA binding transcription factor activity	12	5.13 × 10^−05^	0.00096
6	Transcription from RNA polymerase II promoter	81	6.07 × 10^−05^	0.00096
7	Enzyme binding	56	6.74 × 10^−05^	0.00096
8	DNA binding	107	8.38 × 10^−05^	0.00096
9	Phosphatase binding	12	8.64 × 10^−05^	0.00096
10	Chromatin binding	22	0.000184	0.001608
**D. KEGG Pathways Regulated by Downregulated miR**				
**S. No.**	**Name**	**Hits**	***p*-Value**	**Adj. *p*-Value**
1	Pathways in cancer	64	1.45 × 10^−18^	1.45 × 10^−16^
2	Prostate cancer	31	2.65 × 10^−16^	1.33 × 10^−14^
3	Chronic myeloid leukemia	26	7.80 × 10^−14^	2.60 × 10^−12^
4	Small cell lung cancer	26	9.30 × 10^−13^	2.33 × 10^−11^
5	Glioma	22	2.27 × 10^−11^	4.54 × 10^−10^
6	P53 signaling pathway	22	6.30 × 10^−11^	1.05 × 10^−09^
7	Pancreatic cancer	22	8.72 × 10^−11^	1.25 × 10^−09^
8	Cell cycle	29	3.71 × 10^−10^	4.64 × 10^−09^
9	HTLV-I infection	37	1.41 × 10^−09^	1.57 × 10^−08^
10	Melanoma	20	3.22 × 10^−09^	3.22 × 10^−08^
**E. Geneontology Enrichment for Biological Process (GO-BP) Regulated by Downregulated miR**				
**S. No.**	**Name**	**Hits**	***p*-Value**	**Adj. *p*-Value**
1	Negative regulation of transcription from RNA polymerase II promoter	74	2.97 × 10^−11^	2.97 × 10^−09^
2	Interphase of mitotic cell cycle	59	2.11 × 10^−09^	1.06 × 10^−07^
3	Interphase	59	4.22 × 10^−09^	1.41 × 10^−07^
4	Negative regulation of RNA metabolic process	105	1.76 × 10^−08^	3.22 × 10^−07^
5	Negative regulation of transcription, DNA-dependent	102	1.93 × 10^−08^	3.22 × 10^−07^
6	Negative regulation of cellular metabolic process	102	1.93 × 10^−08^	3.22 × 10^−07^
7	Regulation of cell cycle	151	4.11 × 10^−08^	5.23 × 10^−07^
8	Regulation of transcription from RNA polymerase II promoter	93	4.18 × 10^−08^	5.23 × 10^−07^
9	Negative regulation of cellular biosynthetic process	146	7.03 × 10^−08^	7.81 × 10^−07^
10	Negative regulation of metabolic process	117	1.14 × 10^−07^	1.14 × 10^−06^
**F. Gene Ontology Enrichment for Molecular Function (GO-MF) Regulated by Downregulated miR**				
**S. No.**	**Name**	**Hits**	***p*-Value**	**Adj. *p*-Value**
1	Negative regulation of transcription, DNA-dependent	102	1.72 × 10^−09^	1.72 × 10^−07^
2	Transcription from RNA polymerase II promoter	167	6.75 × 10^−09^	2.40 × 10^−07^
3	Transcription factor binding	62	9.59 × 10^−09^	2.40 × 10^−07^
4	Enzyme binding	115	1.11 × 10^−08^	2.40 × 10^−07^
5	Kinase binding	54	1.20 × 10^−08^	2.40 × 10^−07^
6	Protein kinase binding	48	1.14 × 10^−07^	1.9 × 10^−06^
7	Positive regulation of transcription, DNA-dependent	111	1.5 × 10^−06^	2.14 × 10^−05^
8	Nucleotide binding	191	2.08 × 10^−06^	0.000026
9	SMAD binding	15	4.8 × 10^−06^	5.33 × 10^−05^
10	Phosphatase binding	20	7.74 × 10^−06^	7.74 × 10^−05^

Pathway enrichment analysis for deregulated miR in ovarian cancer: Gene targets for miR were predicted using miRNet web tool (https://www.mirnet.ca/) and miRTarBase v8.0. (**A**,**D**) KEGG pathway analysis enrichment, gene ontology enrichment for (**B**,**E**) biological process (GO-BP) and (**C**,**F**) molecular function (GO-MF) were analyzed for upregulated (**A**–**C**) and downregulated (**D**–**F**) miR target genes; the 10 most significant pathways based on the network generated using degree cutoff 5 are shown.

**Table 3 cancers-13-01085-t003:** Deregulated miR in Endometrial Cancer.

Related Entity	Lit Str.	Lit MIM	Regulation	No. of Papers	PMID
miR-205	37	0.1	Up	27	[300]
miR-200C	28.7	−0.1	Up	20	[309,310]
miR-200A	16.3	0.2	Up	18	[198]
miR-200B	16.3	0.1	Up	18	[311]
miR-21	27.6	−2.4	Up	14	[312]
miR-155	19.2	−2.3	Up	11	[313]
miR-141	9	−0.4	Up	9	[314]
miR-182	14.1	−0.5	Up	9	[315]
miR-135B	7.6	−0.3	Up	5	[316]
miR-183	11.3	−0.8	Up	5	[317]
miR-106B	4.7	−1	Up	4	[318]
miR-10B	4.7	−1.3	Up	4	[319]
miR-210	5.8	−2.6	Up	4	[320]
miR-191	5	−0.9	Up	3	[321]
miR-223	4.4	−1.8	Up	3	[322]
miR-522	4.5	0.5	Up	1	[323]
miR-544A	6.1	0.2	Up	1	[323]
miR-940	4.5	−0.9	Up	1	[324]
miR-944	9.3	−0.3	Up	1	[325]
miR-152	17.3	0.2	Down	10	[326]
miR-145	15.5	−1.8	Down	9	[327]
miR-204	9.1	−0.6	Down	8	[328]
miR-143	7.6	−1.7	Down	6	[329]
miR-203A	13.4	−1.3	Down	6	[330]
miR-424	27.2	0.2	Down	6	[331]
miR-214	4.2	−2.3	Down	5	[332]
miR-23B	4.7	−1	Down	5	[333]
miR-130B	7.3	−0.3	Down	5	[334]
miR-126	7.9	−2.3	Down	4	[335]
miR-195	7.9	−1.7	Down	4	[336]
miR-137	5	−2.1	Down	3	[337]
miR-142	6.3	−0.3	Down	3	[338,339]
miR-181C	4.2	−0.8	Down	3	[340]
miR-23A	7.4	−1.9	Down	3	[341]
miR-29C	5.8	−1.6	Down	3	[342]
miR-320A	7.4	−1.2	Down	3	[343]
miR-146A	4.2	−3.3	Down	3	[344]
miR-1271	13.5	0.2	Down	2	[345]
miR-129-2	8.6	1.5	Down	2	[346]
miR-148B	8.2	−0.7	Down	2	[347]
miR-202	6.9	−1	Down	2	[348]
miR-206	5.8	−2.4	Down	2	[349]
miR-646	9.8	0.6	Down	2	[350]
miR-125A	4.5	−2.1	Down	1	[351]
miR-23C	4.2	1.4	Down	1	[352]
miR-873	10.1	−0.6	Down	1	[353]

Deregulated miR in endometrial cancer. Lit Str. and Lit MIM represent: literature strength, and literature mutual information, respectively. The miR were regrouped based on their expression. The number of publications with the miR was retrieved from PubMed, and representative manuscripts are cited.

**Table 4 cancers-13-01085-t004:** Altered Pathways by Deregulated miR in Endometrial Cancer.

A. KEGG Pathways Regulated by Upregulated miR				
S. No.	Name	Hits	*p*-Value	Adj. *p*-Value
1	Pathways in cancer	9	3.27 × 10^−08^	1.3407 × 10^−06^
2	Prostate cancer	5	0.00000347	0.000071135
3	Glioma	4	0.0000306	0.00037515
4	p53 signaling pathway	4	0.0000366	0.00037515
5	Melanoma	3	0.000952	0.006792333
6	Pancreatic cancer	3	0.000994	0.006792333
7	Small cell lung cancer	3	0.00153	0.008961429
8	Focal adhesion	4	0.00231	0.01183875
9	Bladder cancer	2	0.00322	0.01466889
10	mTOR signaling pathway	2	0.00765	0.031365
**B. Gene Ontology Enrichment for Biological Process (GO-BP) Regulated by Upregulated miR**				
**S. No.**	**Name**	**Hits**	***p*-Value**	**Adj. *p*-Value**
1	negative regulation of apoptotic process	12	1.01 × 10^−10^	4.10 × 10^−09^
2	negative regulation of programmed cell death	12	1.01 × 10^−10^	4.10 × 10^−09^
3	regulation of gene expression	12	1.23 × 10^−10^	4.10 × 10^−09^
4	apoptotic process	21	3.23 × 10^−09^	8.05 × 10^−08^
5	programmed cell death	16	4.83 × 10^−09^	8.05 × 10^−08^
6	regulation of apoptotic process	16	4.83 × 10^−09^	8.05 × 10^−08^
7	regulation of programmed cell death	16	5.83 × 10^−09^	8.33 × 10^−08^
8	cell proliferation	14	8.25 × 10^−09^	1.03 × 10^−07^
9	regulation of RNA metabolic process	14	9.61 × 10^−09^	1.07 × 10^−07^
10	regulation of nucleobase-containing compound metabolic process	15	1.20 × 10^−08^	1.20 × 10^−07^
**C. Gene Ontology Enrichment for Molecular Function (GO-MF) Regulated by Upregulated miR**				
**S. No.**	**Name**	**Hits**	***p*-Value**	**Adj. *p*-Value**
1	negative regulation of transcription, DNA-dependent	10	0.00000168	0.00016128
2	transcription from RNA polymerase II promoter	12	0.0000173	0.0008304
3	protein kinase binding	5	0.000325	0.0101376
4	enzyme binding	8	0.000476	0.0101376
5	kinase binding	5	0.000528	0.0101376
6	RNA polymerase II distal enhancer sequence-specific DNA binding transcription factor activity	3	0.000768	0.012288
7	sequence-specific DNA binding	6	0.001	0.01365333
8	ubiquitin-protein ligase activity	4	0.00124	0.01365333
9	transcription factor binding	5	0.00128	0.01365333
10	small conjugating protein ligase activity	4	0.00162	0.01413818
**D. KEGG Pathways Regulated by Downregulated miR**				
**S. No.**	**Name**	**Hits**	***p*-Value**	**Adj. *p*-Value**
1	Chronic myeloid leukemia	11	9.87 × 10^−12^	9.57 × 10^−10^
2	Pathways in cancer	17	1.30 × 10^−10^	6.31 × 10^−09^
3	Colorectal cancer	8	4.76 × 10^−09^	1.54 × 10^−07^
4	Glioma	8	4.83 × 10^−08^	1.17 × 10^−06^
5	Melanoma	8	6.94 × 10^−08^	0.000001261
6	Pancreatic cancer	8	7.80 × 10^−08^	0.000001261
7	Prostate cancer	8	4.88 × 10^−07^	6.22 × 10^−06^
8	Focal adhesion	11	5.13 × 10^−07^	6.22 × 10^−06^
9	Thyroid cancer	5	0.000003	3.23 × 10^−05^
10	ErbB signaling pathway	7	0.00000682	0.000066154
**E. Gene Ontology Enrichment for Biological Process (GO-BP) Regulated by Downregulated miR**				
**S. No.**	**Name**	**Hits**	***p*-Value**	**Adj. *p*-Value**
1	regulation of cell proliferation	27	5.61 × 10^−07^	0.000023725
2	positive regulation of metabolic process	39	8.03 × 10^−07^	0.000023725
3	regulation of cellular protein metabolic process	28	9.08 × 10^−07^	0.000023725
4	gland development	12	9.49 × 10^−07^	0.000023725
5	tissue morphogenesis	16	0.0000012	0.000024
6	morphogenesis of an epithelium	14	0.00000152	2.444 × 10^−05^
7	negative regulation of cell proliferation	16	0.00000185	2.444 × 10^−05^
8	negative regulation of metabolic process	30	0.00000206	2.444 × 10^−05^
9	regulation of kinase activity	18	0.0000022	2.444 × 10^−05^
10	enzyme linked receptor protein signaling pathway	23	0.00000271	2.692 × 10^−05^
**F. Gene Ontology Enrichment for Molecular Function (GO-MF) Regulated by Downregulated miR**				
**S. No.**	**Name**	**Hits**	***p*-Value**	**Adj. *p*-Value**
1	protein complex binding	10	0.0000896	0.00896
2	transcription from RNA polymerase II promoter	25	0.000725	0.02716667
3	negative regulation of transcription, DNA-dependent	16	0.000815	0.02716667
4	SMAD binding	4	0.00111	0.02775
5	kinase binding	9	0.00194	0.0388
6	protein kinase binding	8	0.00372	0.062
7	SH3/SH2 adaptor activity	3	0.00508	0.07257143
8	transcription factor binding	9	0.00705	0.085
9	phosphatase binding	4	0.00765	0.085
10	cytokine receptor binding	6	0.00892	0.0892

Pathway enrichment analysis for deregulated miR in endometrial cancer: Gene targets for miR were predicted using the miRNet web tool (https://www.mirnet.ca/) and miRTarBase v8.0. (**A**,**D**) KEGG pathway analysis enrichment, gene ontology enrichment for (**B**,**E**) biological process (GO-BP) and (**C**,**F**) molecular function (GO-MF) were analyzed for upregulated (**A**–**C**) and downregulated (**D**–**F**) miR target genes, the 10 most significant pathways based on the network generated using degree cutoff 5 are shown.

**Table 5 cancers-13-01085-t005:** Deregulated miR in Cervical Cancer.

Related Entity	Lit Str.	Lit MIM	Regulation	No. of Papers	PMID
miR-21	136.6	−1.6	Up	61	[383]
miR-205	31.8	−0.9	Up	20	[384]
miR-155	36.1	−2.5	Up	19	[383]
miR-20A	27.8	−0.6	Up	17	[356]
miR-126	30	−1.6	Up	12	[385]
miR-200B	16.5	−1.1	Up	9	[386]
miR-150	10.5	−1.9	Up	8	[363]
miR-200A	9.4	−1.4	Up	8	[387]
miR-944	13.7	0.6	Up	7	[388]
miR-130A	20.6	−1	Up	6	[389]
miR-106B	18.4	−0.7	Up	6	[390]
miR-27B	12.9	−1.3	Up	6	[391]
miR-15B	7.8	−0.9	Up	6	[392]
miR-133B	7.6	−1.6	Up	6	[393]
miR-18A	13.2	−1.6	Up	5	[394]
miR-106A	11.8	−0.9	Up	5	[357]
miR-499A	9.8	−0.7	Up	4	[395]
miR-152	9.5	−1.6	Up	4	[396]
miR-135B	8.7	−1.3	Up	4	[397]
miR-210	6.8	−2.5	Up	4	[398]
miR-940	5.5	−0.5	Up	4	[399]
miR-149	4.1	−1	Up	4	[400]
miR-130B	7.1	−1.4	Up	3	[401]
miR-1290	3.9	−0.6	Up	3	[402]
miR-373	13.8	−1.5	Up	2	[403]
miR-492	7.4	−0.3	Up	2	[236]
miR-205HG	5.8	0.9	Up	2	[404]
miR-127	4.2	−1.1	Up	2	[405]
miR-92B	3.7	−1.6	Up	2	[406]
miR-613	3.4	−1.3	Up	2	[407]
miR-301B	7.4	0	Up	1	[408]
miR-501	6.9	0	Up	1	[409]
miR-3142	5.3	1.4	Up	1	[410]
miR-519D	5.3	−1.2	Up	1	[411]
miR-629	4.5	−0.5	Up	1	[412]
miR-421	4.2	−1.2	Up	1	[413]
miR-145	64.5	−1.2	Down	29	[414]
miR-143	44	−0.6	Down	28	[150]
miR-146A	27.4	−1.7	Down	22	[415]
miR-203A	25	−0.9	Down	22	[416]
miR-214	44.8	−0.9	Down	20	[417]
miR-195	40.1	−0.9	Down	14	[418]
miR-29A	23.6	−1.7	Down	13	[419]
miR-424	18.4	−0.4	Down	12	[420]
miR-7	11	−2.1	Down	12	[421]
miR-206	19.2	−1.7	Down	10	[422]
miR-22	25.3	−1.2	Down	9	[423]
miR-23B	21.4	−1	Down	9	[424]
miR-182	18.7	−1.5	Down	9	[425]
miR-497	16.3	−0.9	Down	8	[426]
miR-183	10	−1.7	Down	7	[427]
miR-125A	10	−1.4	Down	6	[428]
miR-204	14.8	−1.9	Down	5	[429]
miR-144	13.7	−1.4	Down	5	[359]
miR-506	12.9	−0.6	Down	5	[430]
miR-187	10.8	−0.2	Down	5	[431]
miR-132	8.1	−2	Down	5	[432]
miR-223	6	−2.5	Down	5	[433]
miR-215	6.3	−1	Down	4	[434]
miR-216B	4.5	−1.8	Down	4	[435]
miR-10B	14.3	−2	Down	3	[436]
miR-26B	13	−2.1	Down	3	[437]
miR-217	12.7	−1.6	Down	3	[438]
miR-432	11.9	0.5	Down	3	[439]
miR-641	11.1	0.5	Down	3	[440]
miR-744	10	−0.1	Down	3	[441]
miR-15A	8.6	−1.5	Down	3	[442]
miR-200C	6.3	−2.6	Down	3	[443]
miR-186	5.5	−1.2	Down	3	[444]
miR-30A	5.5	−2	Down	3	[445]
miR-142	5	−1.4	Down	3	[446]
miR-196B	4.2	−1.3	Down	3	[447]
miR-4429	4.2	1.5	Down	3	[448]
miR-1284	8.2	0.4	Down	2	[449]
miR-612	6.9	−0.9	Down	2	[450]
miR-873	6.6	−0.6	Down	2	[451]
miR-504	6.1	−1.1	Down	2	[452]
miR-802	6.1	−1.4	Down	2	[453]
miR-99B	6.1	−1.3	Down	2	[454]
miR-320A	6	−1.2	Down	2	[289]
miR-374B	5.8	−0.3	Down	2	[455]
miR-1297	5	−0.4	Down	2	[456]
miR-2861	5	0.2	Down	2	[457]
miR-760	5	−0.7	Down	2	[458]
miR-136	4.2	−1	Down	2	[459]
miR-498	4.2	−0.8	Down	2	[460]
miR-758	4.2	0.3	Down	2	[461]
miR-372	3.4	−1	Down	2	[462]
miR-137	3.1	−2.4	Down	2	[463]
miR-148A	8.7	−2.1	Down	1	[464]
miR-877	7.7	−0.2	Down	1	[465]
miR-889	6.1	−0.2	Down	1	[466]
miR-1202	5.3	−0.3	Down	1	[467]
miR-1294	5.3	−0.8	Down	1	[468]
miR-3148	5.3	1.1	Down	1	[469]
miR-636	5.3	0.1	Down	1	[470]
miR-376C	4.5	−0.9	Down	1	[471]
miR-584	4.5	−0.6	Down	1	[472]
miR-LET7B	3.9	−1.8	Down	1	[473]
miR-134	3.7	−2.6	Down	1	[474]
miR-147A	3.7	0.4	Down	1	[475]
miR-197	3.7	−1.6	Down	1	[476]
miR-6893	3.7	2.4	Down	1	[477]
miR-8075	3.7	2.4	Down	1	[478]
miR-140	3.1	−1.5	Down	1	[479]
miR-630	7.4	−0.9	Down	1	[480]

Deregulated miR in cervical cancer. Lit Str. and Lit MIM represent: literature strength, and literature mutual information, respectively. The miR were regrouped based on their expression. The number of publications with the miR was retrieved from PubMed, and representative manuscripts are cited.

**Table 6 cancers-13-01085-t006:** Altered Pathways by Deregulated miR in Cervical Cancer.

A. KEGG Pathways Regulated by Upregulated miR				
S. No.	Name	Hits	*p*-Value	Adj. *p*-Value
1	p53 signaling pathway	16	1.29 × 10^−09^	1.29 × 10^−07^
2	Bladder cancer	10	3.39 × 10^−08^	1.2867 × 10^−06^
3	Pathways in cancer	32	3.86 × 10^−08^	1.2867 × 10^−06^
4	Chronic myeloid leukemia	14	2.33 × 10^−07^	5.825 × 10^−06^
5	Prostate cancer	15	3.67 × 10^−07^	0.0000064
6	Glioma	13	3.84 × 10^−07^	0.0000064
7	Pancreatic cancer	12	0.0000052	7.4286 × 10^−05^
8	Cell cycle	16	0.00000806	0.00010075
9	Small cell lung cancer	12	0.0000251	0.000266
10	Melanoma	11	0.0000266	0.000266
**B. Gene Ontology Enrichment for Biological Process (GO-BP) Regulated by Upregulated miR**				
**S. No.**	**Name**	**Hits**	***p*-Value**	**Adj. *p*-Value**
1	negative regulation of transcription from RNA polymerase II promoter	49	1.12 × 10^−08^	0.00000112
2	DNA-dependent transcription, initiation	26	0.00000331	0.000096
3	regulation of transcription from RNA polymerase II promoter	93	0.00000641	0.000096
4	transcription initiation from RNA polymerase II promoter	23	0.00000653	0.000096
5	G1 phase of mitotic cell cycle	10	0.00000658	0.000096
6	negative regulation of transcription, DNA-dependent	64	0.00000693	0.000096
7	negative regulation of cellular biosynthetic process	64	0.00000693	0.000096
8	regulation of gene expression	75	0.00000768	0.000096
9	G1 phase	212	0.00000867	9.6333 × 10^−05^
10	regulation of translation	10	0.00000977	0.0000977
**C. Gene Ontology Enrichment for Molecular Function (GO-MF) Regulated by Upregulated miR**				
**S. No.**	**Name**	**Hits**	***p*-Value**	**Adj. *p*-Value**
1	negative regulation of transcription, DNA-dependent	64	0.00000398	0.000398
2	DNA binding	140	0.00000949	0.0004745
3	transcription from RNA polymerase II promoter	102	0.0000457	0.0012075
4	transcription factor binding	37	0.0000483	0.0012075
5	phosphatase binding	14	0.000079	0.00158
6	sequence-specific DNA binding	46	0.000203	0.00338333
7	enzyme binding	67	0.000247	0.00352857
8	double-stranded DNA binding	15	0.000344	0.0043
9	structure-specific DNA binding	20	0.000544	0.00604444
10	RNA polymerase II distal enhancer sequence-specific DNA binding transcription factor activity	12	0.000643	0.00643
**D. KEGG Pathways Regulated by Downregulated miR**				
**S. No.**	**Name**	**Hits**	***p*-Value**	**Adj. *p*-Value**
1	Pathways in cancer	61	7.79 × 10^−17^	7.79 × 10^−15^
2	Prostate cancer	30	1.77 × 10^−15^	8.85 × 10^−14^
3	Cell cycle	34	5.07 × 10^−14^	1.69 × 10^−12^
4	Colorectal cancer	20	2.79 × 10^−12^	6.98 × 10^−11^
5	Chronic myeloid leukemia	23	3.41 × 10^−11^	6.82 × 10^−10^
6	Pancreatic cancer	22	7.14 × 10^−11^	1.19 × 10^−09^
7	Glioma	21	1.48 × 10^−10^	2.11 × 10^−09^
8	Neurotrophin signaling pathway	29	2.36 × 10^−10^	2.95 × 10^−09^
9	Focal adhesion	38	3.17 × 10^−10^	3.52 × 10^−09^
10	Endometrial cancer	17	3.62 × 10^−10^	3.62 × 10^−09^
**E. Gene Ontology Enrichment for Biological Process (GO-BP) Regulated by Downregulated miR**				
**S. No.**	**Name**	**Hits**	***p*-Value**	**Adj. *p*-Value**
1	regulation of cellular protein metabolic process	141	1.46 × 10^−10^	6.50 × 10^−09^
2	negative regulation of programmed cell death	78	2.45 × 10^−10^	6.50 × 10^−09^
3	negative regulation of apoptotic process	77	2.60 × 10^−10^	6.50 × 10^−09^
4	regulation of cell cycle	77	2.60 × 10^−10^	6.50 × 10^−09^
5	negative regulation of RNA metabolic process	91	1.04 × 10^−09^	1.51 × 10^−08^
6	negative regulation of cellular biosynthetic process	101	1.06 × 10^−09^	1.51 × 10^−08^
7	negative regulation of transcription, DNA-dependent	115	1.06 × 10^−09^	1.51 × 10^−08^
8	negative regulation of biosynthetic process	98	1.36 × 10^−09^	1.51 × 10^−08^
9	regulation of translation	98	1.36 × 10^−09^	1.51 × 10^−08^
10	gland development	115	2.75 × 10^−09^	2.75 × 10^−08^
**F. Gene Ontology Enrichment for Molecular Function (GO-MF) Regulated by Downregulated miR**				
**S. No.**	**Name**	**Hits**	***p*-Value**	**Adj. *p*-Value**
1	negative regulation of transcription, DNA-dependent	98	9.93 × 10^−11^	9.93 × 10^−09^
2	enzyme binding	110	7.08 × 10^−10^	3.54 × 10^−08^
3	SMAD binding	18	5.26 × 10^−09^	1.59 × 10^−07^
4	kinase binding	51	6.34 × 10^−09^	1.59 × 10^−07^
5	sequence-specific DNA binding	73	2.57 × 10^−08^	5.14 × 10^−07^
6	protein kinase binding	46	3.29 × 10^−08^	5.48 × 10^−07^
7	nucleotide binding	182	5.93 × 10^−08^	8.47 × 10^−07^
8	positive regulation of transcription, DNA-dependent	106	1.40 × 10^−07^	0.00000175
9	transcription from RNA polymerase II promoter	146	3.75 × 10^−07^	4.1667 × 10^−06^
10	transcription factor binding	53	6.16 × 10^−07^	0.00000616

Pathway enrichment analysis for deregulated miR in cervical cancer: Gene targets for miR were predicted using the miRNet web tool (https://www.mirnet.ca/) and miRTarBase v8.0. (**A**,**D**) KEGG pathway analysis enrichment, gene ontology enrichment for (**B**,**E**) biological process (GO-BP), and (**C**,**F**) molecular function (GO-MF) were analyzed for upregulated (**A**–**C**) and downregulated (**D**–**F**) miR target genes, the 10 most significant pathways based on the network generated using degree cutoff 5 are shown.

## Data Availability

Available on request.

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
