# Peer review of "Small Non-Coding-RNA in Gynecological Malignancies"

_cancers, 2021, doi:10.3390/cancers13051085_

Round 1

Reviewer 1 Report

I appreciate Authors' work and the time spent on literature analysis, however, in my opinion authors reach opposing effect revealed by introduction of too many references.

This review summarizes many literature reports, however, some data are difficult to read and understand by readers, such as tables, which are confusing rather than informative.

I suggest authors to reduce paper length and rather focus on the clinical aspects and to be more precise and readable. 

Author Response

Reviewer #1

Comment: 1

I appreciate Authors' work and the time spent on literature analysis, however, in my opinion authors reach opposing effect revealed by introduction of too many references.

Response:

We thank the reviewer for the encouraging comments. We have revised the manuscript by reducing table length and images by increasing the degree cutoff for better representation and visibility. Where possible, we have also removed references and reduced them by 54. However, in order to provide a comprehensive review of the literature, we feel it is important to discuss the opposing effects when reported.

Comment: 2

This review summarizes many literature reports, however, some data are difficult to read and understand by readers, such as tables, which are confusing rather than informative.

Response:

We appreciate the suggestion, please refer to response1.

Comment: 3

I suggest authors to reduce paper length and rather focus on the clinical aspects and to be more precise and readable.

Response:

Please refer to response 1.

Reviewer 2 Report

In this manuscript, Dwivedi et al. comprehensively review the role of non-coding RNAs in gynecological malignancies. This article is fairly well written and covers important aspects of non-coding biology in these cancers. 

My main suggestion would be to reduce table 2, 4 and 6 to the top 5 or 10 hits to shorten the manuscript and make it more concise (I believe the list of miRNAs is fine as it is).

Minor Comments

  • The format of each heading is confusing - some are in italics and some are not. Please make this consistent through.
  • Section 2.3 - miRNAs are not often referred to as Micro-ribonucleic acids (miR), it would be better to call them miRNAs.

Author Response

Comment:

In this manuscript, Dwivedi et al. comprehensively review the role of non-coding RNAs in gynecological malignancies. This article is fairly well written and covers important aspects of non-coding biology in these cancers. 

Response:

We thank the reviewer for the encouraging comments.

Comment:

My main suggestion would be to reduce tables 2, 4, and 6 to the top 5 or 10 hits to shorten the manuscript and make it more concise (I believe the list of miRNAs is fine as it is).

Response:

In agreement, we have revised the manuscript by reducing table length and images by increasing degree cutoff for better representation and visibility.

Minor Comments

  • The format of each heading is confusing - some are in italics and some are not. Please make this consistent through.

Response:

We thank the reviewer for pointing this out, formatting has been revised accordingly.

  • Section 2.3 - miRNAs are not often referred to as Micro-ribonucleic acids (miR), it would be better to call them miRNAs.

Response:

We thank the reviewer for pointing this out, it has been revised in the manuscript as suggested.

Reviewer 3 Report

The authors summarized the importance of small non-coding-RNA as biomarkers in different gynaecological malignancies, and potential usage for diagnosis and therapy. This review provides a great review of piRNA, miRNA, tRNA-derived small RNA in various gynaecological malignancies.

Line 111: The authors wrote: „Of these, epithelial ovarian cancer (EOC) is the most common, representing 85–90% of ovarian cancers [22].” However the Ref 22 refers to endometrial carcinoma “22. McAlpine, J.; Leon-Castillo, A.; Bosse, T. The rise of a novel classification system for endometrial carcinoma; integration of molecular subclasses. J Pathol 2018, 244, 538-549, doi:10.1002/path.5034.”

Line 129: Not just the CA125, but also the Wnt, HE4, p53 as reliable biomarkers are used in histopathological differential diagnosis of ovarian cancer.

Figure 5 and 7: the figures are cramped. It is hard to construe due to the large amount of data.

Minor points:

The manuscript is well written, there are only a few mistakes, and it requires a minor spell check. e.g. lines 186, 406, or line 350: “[120-122] [123]”, the authors should revise the references in the text.

Author Response

Comment:

The authors summarized the importance of small non-coding-RNA as biomarkers in different gynaecological malignancies, and potential usage for diagnosis and therapy. This review provides a great review of piRNA, miRNA, tRNA-derived small RNA in various gynaecological malignancies.

Response:

We thank the reviewer for the encouraging comments.

Comment:

Line 111: The authors wrote: „Of these, epithelial ovarian cancer (EOC) is the most common, representing 85–90% of ovarian cancers [22].” However the Ref 22 refers to endometrial carcinoma “22. McAlpine, J.; Leon-Castillo, A.; Bosse, T. The rise of a novel classification system for endometrial carcinoma; integration of molecular subclasses. J Pathol 2018, 244, 538-549, doi:10.1002/path.5034.”

Response:

We apologize for such errors and the above section was revised with appropriate citation.

Comment:

Line 129: Not just the CA125, but also the Wnt, HE4, p53 as reliable biomarkers are used in histopathological differential diagnosis of ovarian cancer.

Response:

We appreciate the reviewer’s suggestions, the above section is revised with appropriate literature and citations.

Comment:

Figure 5 and 7: the figures are cramped. It is hard to construe due to the large amount of data.

Response: We appreciate the suggestion and have revised the manuscript by reducing table length and images by increasing degree cutoff for better representation and visibility.

Minor points:

The manuscript is well written, there are only a few mistakes, and it requires a minor spell check. e.g. lines 186, 406, or line 350: “[120-122] [123]”, the authors should revise the references in the text.

Response:

We thank the reviewer for pointing this out, the manuscript is revised as suggested.

Round 2

Reviewer 1 Report

I have been not convinced of the response and I consider that more efforts should be made to reconsider paper acceptance.